# Analyzing inter-reader variability affecting deep ensemble learning for COVID-19 detection in chest radiographs

**Sivaramakrishnan Rajaraman**[1]*, **Sudhir Sornapudi**[2], **Philip O. Alderson**[3], **Les R. Folio**[4], **Sameer K. Antani**[1]

**1** Lister Hill National Center for Biomedical Communications, National Library of Medicine, Bethesda, Maryland, United States of America, **2** Department of Electrical and Computer Engineering, Missouri University of Science and Technology, Rolla, Missouri, United States of America, **3** School of Medicine, Saint Louis University, St. Louis, Missouri, United States of America, **4** Radiology and Imaging Sciences, Clinical Center, National Institutes of Health, Bethesda, Maryland, United States of America

* sivaramakrishnan.rajaraman@nih.gov

**Data Availability Statement:** All relevant data are within the manuscript.

**Funding:** This study is supported by the Intramural Research Program (IRP) of the National Library of

## Abstract

Data-driven deep learning (DL) methods using convolutional neural networks (CNNs) demonstrate promising performance in natural image computer vision tasks. However, their use in medical computer vision tasks faces several limitations, viz., (i) adapting to visual characteristics that are unlike natural images; (ii) modeling random noise during training due to stochastic optimization and backpropagation-based learning strategy; (iii) challenges in explaining DL black-box behavior to support clinical decision-making; and (iv) inter-reader variability in the ground truth (GT) annotations affecting learning and evaluation. This study proposes a systematic approach to address these limitations through application to the pandemic-caused need for Coronavirus disease 2019 (COVID-19) detection using chest X-rays (CXRs). Specifically, our contribution highlights significant benefits obtained through (i) pretraining specific to CXRs in transferring and fine-tuning the learned knowledge toward improving COVID-19 detection performance; (ii) using ensembles of the fine-tuned models to further improve performance over individual constituent models; (iii) performing statistical analyses at various learning stages for validating results; (iv) interpreting learned individual and ensemble model behavior through class-selective relevance mapping (CRM)-based region of interest (ROI) localization; and, (v) analyzing inter-reader variability and ensemble localization performance using Simultaneous Truth and Performance Level Estimation (STAPLE) methods. We find that ensemble approaches markedly improved classification and localization performance, and that inter-reader variability and performance level assessment helps guide algorithm design and parameter optimization. To the best of our knowledge, this is the first study to construct ensembles, perform ensemble-based disease ROI localization, and analyze inter-reader variability and algorithm performance for COVID-19 detection in CXRs.

Medicine (NLM) and the National Institutes of Health (NIH). The intramural research scientists (authors) at the NIH dictated study design, data collection, data analysis, decision to publish and preparation of the manuscript.

**Competing interests:** The authors have declared that no competing interests exist.

## Introduction

Coronavirus disease 2019 (COVID-19) is caused by the new Severe Acute Respiratory Syndrome Coronavirus 2 (SARS-CoV-2) that originated in Wuhan, China. The World Health Organization (WHO) declared this disease spread as an ongoing pandemic [1]. As of July 2020, the pandemic has resulted in over 14 million cases, and more than 600,000 deaths worldwide. The disease commonly infects the lungs and results in pneumonia-like symptoms [2]. Reverse transcription-polymerase chain reaction (RT-PCR) analysis is the gold standard to confirm infections. However, these tests are reported to exhibit varying sensitivity and are not widely available [2]. Radiological imaging using chest X-rays (CXRs) and computed tomography (CT) scans, though not currently recommended in the United States, are commonly used radiological diagnostic support aids to manage COVID-19 disease progression [2]. While CT scans are more sensitive in detecting pulmonary disease manifestations than CXRs, their use is limited due to issues such as non-portability, repeated sanitation requirements for CT examination rooms and equipment, and exposing patients, hospital staff and technical personnel to the infection. According to the American College of Radiology (ACR) recommendations [3], CXRs are considered a viable alternative to CT scans in addressing some of these limitations. However, the pandemic nature of the disease has compounded the existing shortage of expert radiologists, particularly in third-world countries [4]. Under these circumstances, artificial intelligence (AI) driven computer-aided diagnostic (CADx) tools have been considered as potentially viable alternatives for facilitating swift patient referrals or aiding appropriate medical care [5]. Several studies using data-driven deep learning (DL) algorithms with convolutional neural network (CNN) models in various strategies have been published for detecting, localizing, or measuring progression of COVID-19 using CXRs and CTs [4, 6, 7]. While there are scores of medical imaging CADx solutions that use DL approaches for disease detection including COVID-19, there are significant limitations in existing approaches related to data set type, size, scope, model architecture, and evaluation. We address these concerns and propose novel analyses to meet the urgent demand for COVID-19 detection using CXRs.

### Modality-specific transfer learning and ensemble learning

Existing solutions tend to be disease-specific and require retraining on a large-collection of expert-annotated data to ensure use in real-world applications. Generalization of these approaches is challenged by available expert-annotations, their strength (i.e. weak image-level labels versus strong region of interest (ROI) localizing the pathology), and necessary computation resources. Under these circumstances, transfer learning strategies are commonly adopted [8] where the models are trained on a large-scale selection of stock photographic images like ImageNet [9] and then fine-tuned for the specific task. A problem with this approach is that the architecture and hyperparameters of these pre-trained models are optimized for natural image computer vision applications. In contrast, medical image collections bearing the desired pathology are significantly smaller in number. Therefore, using these models for medical visual analyses often results in a covariate shift and generalization issues due to the difference in source and target image modalities. Medical images are distinct in their characteristics such as highly localized disease ROIs, and varying appearances for the same disease label and severity [10]. Under these circumstances, the transferred knowledge from the natural image processing domain may not be optimal for disease localization. We propose training deep learning (DL) models with suitable depth on a large-scale selection of medical images of the same modality to learn relevant feature representations that can be transferred and fine-tuned for related medical visual recognition tasks. Such medical modality-specific transfer learning could improve DL performance and generalization by learning the common characteristics of the

source and target modalities. This could lead to a better initialization of model parameters and faster convergence, thereby reducing computational demand, improving efficiency, and increasing opportunity for successful deployment.

Data-driven DL models use non-linear methods and learn through stochastic error back-propagation to perform automated feature extraction and classification. These models scale up in performance by increasing the amount of training data and computational resources. Further, their sensitivity to the training data specifics limits their generalization due to learning different sets of weights at each instance of training. This stochastic learning nature results in different predictions referred to as the variance error. Also, there are issues concerning bias errors due to an oversimplified model that results in predictions that are different from the GT thereby placing a higher demand on appropriate threshold selection for obtaining desired performance. Ensemble learning methods including majority voting, averaging, weighted averaging, stacking, and blending seek to address these issues by combining predictions of multiple models and resulting in a better performance compared to that of any individual constituent model [11].

## ROI localization, observer variability, and statistical analysis

Data-driven medical DL models have often been maligned for their "black box" behavior, i.e., inability to make clear their decision-making process. This is often due to their massive architectural depth resulting in a large number of model parameters and lack of decomposability into individual explainable components. Further, multiple non-linear processing units perform complex data transformations that can result in unpredictable behavior. This results in an apparent opaque relationship between input and predictions which is a serious bottleneck in their use in deriving understandable clinical interpretations.

Supervised learning requires a consistent label associated with the appearance of the pathology in the image. However, in medical images, these labels can vary not only for disease stage and shared appearance with other diseases but also with observer expertise and sensitivity to assessment demands. A new pandemic, for example, may bias experts toward higher sensitivity, i.e. they will tend to associate non-specific features with the new disorder because they lack experience with relevant disease manifestation in the image [1–3]. Therefore, an assessment of observer variability, including analyzing (i) inter-reader, and (ii) intra-reader variability, constitutes an essential part of AI-based classification and localization studies. It is reported that inter-reader variability tends to be higher than intra-reader variability because multiple observers may have a different opinion on the outlining disease-specific ROI depending on their expertise or personal leanings toward recommending necessary clinical care [12]. Thus, inter-reader variability is a major obstacle that may lead to misinterpretation through the "inexact" region of interest (ROI) annotations and also affects supervised learning. Not only can this lead to a false diagnosis or inability to evaluate the true benefit of accurately supplementing clinical-decision making, but it places a greater burden on the number of training images needed to overcome these implicit biases. Thus, it is imperative to conduct inter-reader variability analysis as part of evaluating AI performance. An obvious approach to overcome this challenge might be to compare a collection of annotations by several radiologists using relevant clinical data. However, quantifying expert performance in annotating disease-specific ROI is difficult. This persistent challenge exists because of the difficulty in obtaining or estimating a known true ROI for the task under study. While there exist automated tools to manage inter- and intra-reader variability, these algorithms need to be assessed to warrant their suitability for the task under study. Additionally, it is imperative to determine an appropriate measure for comparing individual expert annotations with each other and with the AI [13].

Results and methods in a study need to be transparently reported to accurately communicate scientific discovery. Statistical analyses are critical for measuring inherent data variability and their impact on AI performance. They help in evaluating claims and differentiating reasonable and uncertain conclusions. Statistical reporting helps to alleviate issues resulting from incorrect data mining, biased samples, overgeneralization, causality, and violating the assumptions concerning analysis. However, a study of the literature reveals that scientific publications are often limited in presenting statistical analyses of their results [14].

In this study, we address the aforementioned limitations through a stage-wise systematic approach, as follows: (i) we explore the benefits of CXR modality-specific pretraining that results in learning CXR modality-specific knowledge, which can be transferred and fine-tuned to improve performance toward COVID-19 detection in CXRs; (ii) we compare the utility of several ImageNet-pretrained CNN models truncated at their empirically determined intermediate layers to that of out-of-the-box ImageNet-pretrained CNNs toward the current task; (iii) we use ensembles of fine-tuned models for COVID-19 detection that are created through various strategies to improve performance compared to any individual constituent model; (iv) we explain learned behavior of individual CNNs and their ensembles using class-selective relevance mapping (CRM)-based localization [15] tools that identify discriminative ROIs involved in detecting COVID-19 viral disease manifestations; (v) we perform ensemble localization to improve localization behavior and compensate for the error due to neglected ROIs by individual CNNs; (vi) we perform exploratory studies to analyze variability in model localization using annotations of two expert radiologists; (vii) we measure statistical significance in performance metrics including Intersection over Union (IoU) and mean average precision (mAP); and, (viii) we perform inter-reader variability analysis using Simultaneous Truth and Performance Level Estimation (STAPLE) [13] that generates a reference consensus annotation from the set of radiologists' annotations. This is compared with individual radiologist annotations and the predicted disease ROI by model ensembles to provide a measure of inter-reader variability and algorithm performance. To our best knowledge, this is the first study to construct ensembles, perform ensemble-based disease ROI localization, and evaluate inter-reader reader variability and algorithm performance toward COVID-19 detection in CXRs.

## Related works

### CXR modality-specific transfer learning and ensemble learning

Yadav et al. [16] demonstrated the benefits of transferring knowledge learned from training on a large-scale selection of CXR images and repurposing them toward tuberculosis (TB) detection. They constructed model ensembles and compared their performance with individual models toward classifying CXRs as showing normal lungs or TB -like manifestations. Rajaraman & Antani [17] proposed CXR modality-specific knowledge transfer by retraining the ImageNet-pretrained CNN models on a large-scale selection of CXRs collected from various institutions. This helped in improving generalization of the learned knowledge that was transferred and fine-tuned to detect TB disease-like manifestations in CXRs. The authors performed ensemble learning using the best-performing CNNs to demonstrate better performance in classifying CXRs as belonging to normal or TB-infected classes. At present, the literature on CXR analysis benefiting from modality-specific knowledge transfer particularly applied to detect COVID-19 viral disease manifestations is limited. This leaves room for progress toward evaluating the efficacy of these methods in improving the performance toward COVID-19 detection. Lakhani & Sundaram [18] used model ensembles to classify CXRs as showing normal lungs or TB-like radiological manifestations. It was observed that an ensemble of custom CNN and ImageNet-pretrained models delivered superior classification

performance with an AUC of 0.99. Rajaraman et al. [19] evaluated the efficacy of a stacked model ensemble constructed from hand-crafted features/classifiers and DL models toward TB detection in CXRs. CXRs collected from various institutions were used to improve the generalization of the proposed approach. It was observed that the model ensembles delivered better performance than individual constituent models in all performance metrics. Ensemble learning has been applied to detect cardiomegaly in CXRs [20]. The authors observed that DL model ensembles were 92% accurate as compared to 76.5% accuracy obtained with hand-crafted features/classifiers. These results demonstrate the superiority of ensemble learning over the traditional approach of evaluating the performance with stand-alone models. Applied to COVID-19 detection in CXRs, Rajaraman et al. [5] iteratively pruned the DL models and constructed ensembles to improve performance compared to individual constituent models. To this end, the authors observed that the weighted average of iteratively pruned models demonstrated superior classification performance with a 99.01% accuracy and AUC of 0.9972. Otherwise, the literature available on applying ensemble learning toward COVID-19 detection in chest radiographs is limited.

## ROI localization, observer variability, and statistical analysis

Exploratory studies in developing explainable and transparent AI solutions toward clinical decision-making are crucial to developing robust solutions for clinical use. Literature studies reveal several works interpreting the learned behavior of DL models by highlighting pixels that impact prediction scores, with varying intensities. Zeiler & Fergus [21] used deconvolution methods to modify the gradients that resulted in qualitatively improving ROI localization. Dosovitskiy & Brox [22] inverted image representations using up-CNN models to provide insights into learned feature representations. Zhou et al. [23] generated class-activation maps (CAM) by mapping the prediction class scores back to the deepest convolutional layer. Selvaraju et al. [24] generalized the use of CAM tools and proposed gradient-weighted CAM (Grad-CAM) methods that can be applied to CNNs with varying architecture. Kim et al. [15] proposed a class-selective relevance mapping (CRM) algorithm to visualize discriminative ROIs in classifying medical image modalities. The authors measured both positive and negative contributions of the feature map spatial elements in the deepest convolutional layer of the trained models toward making class-specific predictions. It was observed that CRM methods delivered superior localization toward classifying medical imaging modalities compared to CAM-based methods. Applied to the task of localizing COVID-19 viral disease manifestations in CXRs and CT scans, Li et al. [7] proposed a DL model called COVNet that learned the underlying feature representations from volumetric CT scans. It was observed that the model showed better performance with an AUC of 0.96 in detecting COVID-19 viral disease patterns and differentiating them from other non-COVID-19 pneumonia-related opacities. They used CAM-based visualization tools to localize the suspicious ROIs toward detecting COVID-19 viral disease manifestations. Karim et al. [25] proposed a custom DL model and used Grad-CAM tools to explain their predictions toward COVID-19 detection. The model achieved a sensitivity of 83% in detecting COVID-19 disease patterns in CXRs. Rajaraman & Antani [6] proposed a weakly-labeled data augmentation approach to increase training data size for recognizing COVID-19 viral related pneumonia opacities in CXRs. They used a strategic approach to train various DL models with non-augmented and weakly-labeled augmented training and evaluated their performance. It was observed that the simple addition of CXRs showing COVID-19 viral disease manifestations to weakly labeled augmented training data improved performance. This study revealed that COVID-19 viral disease patterns have a uniquely different presentation compared to non-COVID-19 viral pneumonia-related opacities. The authors used Grad-

CAM tools to study the behavior of models trained with non-augmented and augmented data toward localizing COVID-19 viral disease manifestations in CXRs. Otherwise, the literature is limited concerning the use of visualization tools toward COVID-19 detection in CXRs. Applied to CXR analysis, Balabanova et al. [26] performed an observational study among Russian clinicians in analyzing the variability toward interpreting abnormalities in CXRs. The agreement was analyzed in different scales using the Kappa statistic for a set of 50 CXRs. It was observed that there existed only a fair agreement in detecting and localizing abnormalities with a Kappa value of 0.380 and 0.448, respectively. This demonstrated that limited agreement on interpreting abnormalities resulted in sub-optimal population screening. Applied to CT scans, Al-Khawari et al. [27] analyzed inter- and intra-radiologist variability in detecting abnormal parenchymal lung manifestations on high-resolution CT scans. They used the Kappa statistic to measure the degree of agreement toward these analyses. A clinically acceptable agreement was observed between the radiologists, but the agreement rate declined when the radiologists were not involved in the regular analysis of thoracic CT scans. Another study [28] analyzed COVID-19 disease manifestations in high-resolution CT scans obtained from patients at the North Sichuan Medical College, Nanchong, China. They assessed inter-observer variability by having CT readers repeat the data analysis at intervals of three days. A comparison of a set of measurements by the same scan reader was used to assess intra-observer variability. They observed the existence of significant variability in inter- and intra-observer analysis, concerning the extent and density of disease spread. At present, there is no available literature on the analysis of inter- and/or intra-reader variability applied to COVID-19 detection in CXRs.

Diong et al. [14] conducted a cross-sectional study toward analyzing the quality of statistical reporting in a random selection of publications in the Journal of Physiology and the British Journal of Pharmacology. The study used samples before and after the publication of an editorial, suggesting measures to adopt in reporting data and statistical analyses. The authors observed no evidence of change in reporting these measures after the editorial publication. They observed that 90–96% of papers were not reporting statistical significance measures including $p$-values to identify the specific groups exhibiting these statistically significant differences in performance. Appropriate statistical analyses are included in the current study.

## Materials and methods

### Data collection

This retrospective study uses the following publicly available datasets:

i. Pediatric CXR dataset: Kermany et al. [29] made available a collection of 5,856 pediatric CXRs showing normal lungs ($n = 1,583$) or bacterial ($n = 2,780$) or viral pneumonia ($n = 1,493$) disease manifestations. The data were collected from children age 1 to 5 years at the Guangzhou Children's Medical Center, China. The radiological examinations were performed as a part of routine clinical care. The CXR images are made available in JPEG format, and approximately 2000 × 2000 pixels resolution with 8-bit depth.

ii. RSNA CXR dataset: Shih et al. [30] made available a collection of 26,684 frontal CXRs for a Kaggle challenge. The CXRs are grouped into to normal ($n = 8,851$) and abnormal ($n = 17,833$) classes; the abnormalities include pneumonia or non-pneumonia related opacities. The CXR images are made available in 1024 × 1024 8-bit pixels resolution and DICOM format.

**Table 1. Demographic study.**

| Dataset | Total | | Mean (age) | | Standard deviation (age) | |
|---|---|---|---|---|---|---|
| | Male | Female | Male | Female | Male | Female |
| NIH [8] | 63340 | 48780 | 47.04 | 46.6 | 17.19 | 16.27 |
| Pediatric CXR [29] | NA | NA | NA | NA | NA | NA |
| RSNA [30] | 17006 | 12888 | NA | NA | NA | NA |
| CheXpert [31] | 132871 | 91007 | 60.83 | 60.43 | 18.19 | 18.19 |
| Montreal-COVID-19 [32] | 131 | 64 | 59.15 | 54.97 | 16.27 | 15.11 |
| Twitter-COVID-19 | 17 | 11 | 13.43 | 19.61 | 8.75 | 8.38 |

The table shows the statistics such as patient count, age, and sex for the various datasets used in this study. NA denotes Not Available.

iii. CheXpert CXR dataset: Irvin et al. [31] made available a collection of 191,219 frontal CXRs showing normal lungs ($n = 17,000$) or other pulmonary abnormalities ($n = 174,219$). The CXR images are collected from patients at Stanford University Hospital, California, and are labeled for various thoracic disease manifestations by an automated natural language processing (NLP)-based labeler. The labels are extracted from radiological texts and conform to the Fleischner Society glossary of terms for thoracic imaging.

iv. NIH CXR-14 dataset: Wang et al. [8] released a collection of 112,120 frontal CXRs, collected from 30,805 patients at the NIH Clinical Center, Maryland. The collection includes CXRs, labeled as showing pulmonary abnormalities ($n = 51,708$) or normal lungs ($n = 60,412$). The CXRs were screened to remove personally identifiable information and ensure patient privacy. The CXRs belonging to the abnormal category are labeled for multiple thoracic disease manifestations using the information extracted from radiological reports using an automated NLP-based labeling algorithm.

v. Twitter-COVID-19 CXR dataset: A radiologist from a hospital in Spain made available a collection of 134 CXRs exhibiting COVID-19 viral pneumonia manifestations, on Twitter (https://twitter.com/ChestImaging). The data were collected from SARS-CoV-2 PCR+ subjects and are made available at approximately 2000 ×2000 pixels resolution.

vi. Montreal-COVID-19 CXR dataset: Cohen et al. [32] manage a GitHub repository that hosts a collection of CXRs and CT scans of SARS-CoV-2 + and/or suspected patients. The images are pooled from publications and hospitals through collaboration with physicians and other public resources. As of May 2020, the collection includes 226 CXRs showing COVID-19 viral pneumonia manifestations. The authors didn't provide complete metadata, however, the collection includes CXRs of 131 male patients and 64 female patients. The demographic information provided by the data providers for the various datasets used in this study are given in Table 1.

## Lung ROI cropping and preprocessing

Input data characteristics directly impact DL model learning, which is significant in applications that involve disease detection. For example, clinical decision-making could be adversely impacted by learning irrelevant features. In the case of COVID-19 and other pulmonary diseases, it is vital to limit analysis to the lung ROI and train the models toward learning relevant feature representations from within these pulmonary zones. Literature studies reveal that U-Net-based semantic segmentation delivers commendable performance in segmentation tasks using natural and medical imagery [33]. For this study, we use a custom U-Net with

dropout [34] layers to segment the lung ROI from the background. Gaussian dropouts are used in the encoder to reduce overfitting and provide restrictive regularization. A dropout ratio of 0.5 is used after empirical pilot evaluations. Fig 1 shows the architecture of the custom U-Net segmentation and its corresponding performance curves. This is the first step in training. The model is trained and validated on patient-specific splits (80/20 train/validation split) of CXRs and their associated lung masks made available by Candemir & Antani [35]. Sigmoidal activation is used at the deepest convolutional layer to restrict the mask pixels to the range (0–1). The model is optimized to minimize a combination of binary cross-entropy and dice losses given by,

$$L_n = w_1 L_{BCE_n} + w_2 L_{DSC_n} \tag{1}$$

where $L_{BCE_n}$ is the binary cross-entropy loss, $L_{DSC_n}$ is the Dice loss, and $n$ is the batch number. The losses are computed for each mini-batch. The final loss for the entire batch is determined

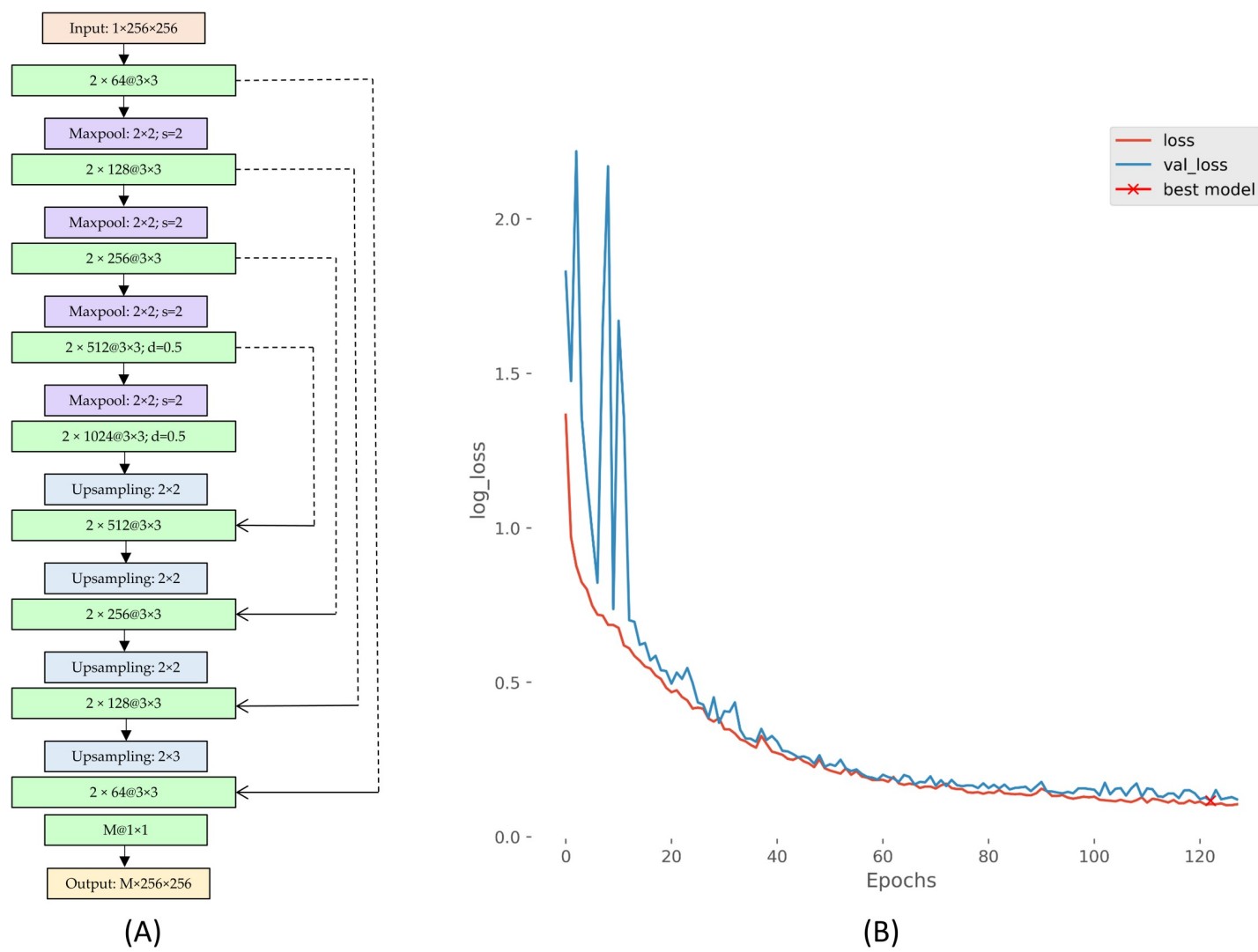

**Fig 1. The architecture of the custom U-Net with dropout and its performance curves.**

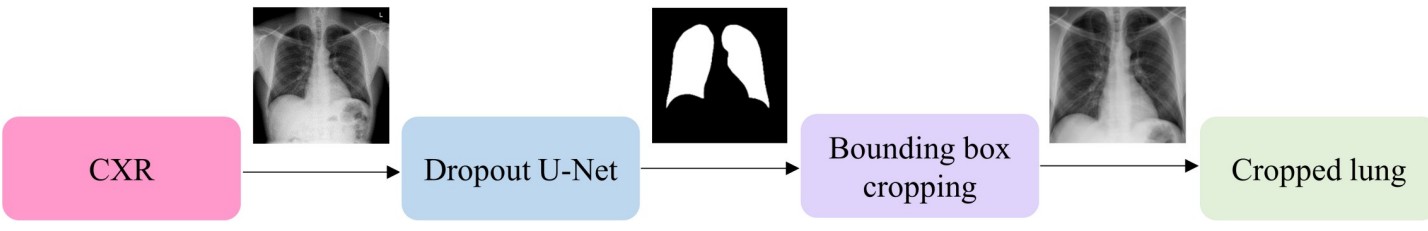

**Fig 2. Segmentation workflow showing UNet-based mask generation and lung ROI cropping.**

by the mean of loss across all the mini-batches. The expression for $L_{BCE_n}$ and $L_{DSC_n}$ is given by:

$$L_{BCE_n} = -[t_n \log(y_n) + (1 - t_n)\log(1 - y_n)] \tag{2}$$

$$L_{DSC_n} = 1 - \frac{2\sum t_n \cdot y_n}{\sum t_n + \sum y_n} \tag{3}$$

where $t$ is the target and $y$ is the output from the final layer. Here, we choose $w_1 = w_2 = 0.5$. Callbacks are used to store model weights after each epoch only when there is a reduction in the validation loss. This helps us select the "best model" at the end of the training phase. The default value of 0.5 is used as the discrimination threshold to convert the predicted probability into the class labels. The best model weights are used for lung mask generation. The model is trained to generate lung masks at $256 \times 256$ pixel resolution for various datasets used in this study. The lung boundaries are delineated using the generated masks and are cropped to a bounding box containing the lung pixels. The lung bounding boxes are resized to $256 \times 256$ pixel dimensions and used for further analysis. The cropped lung bounding boxes are further preprocessed as follows: (i) Images are normalized so that the pixel values are restricted to the range (0–1). (ii) Images are passed through a median filter to perform noise removal and edge preservation. (iii) Image pixels are centered through mean subtraction and are standardized to reduce computational complexity. The segmentation workflow is shown in Fig 2.

## Repeated CXR pretraining and fine-tuning

The steps in training that follow segmentation are shown in Fig 3. First (1), the images are pre-processed to remove irrelevant features by cropping the lung ROI. The cropped images are used for model training and evaluation. We perform repeated CXR-specific pretraining in transferring modality-specific knowledge that is fine-tuned toward detecting COVID-19 viral manifestations in CXRs. To do this, in the next training step (2) the CNNs are trained on a large collection of CXRs to separate normals from those showing abnormalities of any type. Next, (3) we retrain the models from the previous step, focusing on separating CXRs showing bacterial pneumonia or non-COVID-19 viral pneumonia from normals. Next, (4) we fine-tune the models from the previous step toward the specific separation of CXRs showing COVID-19 pneumonia from normals. Finally (5) the learned features from this phase of training become parts of the ensembles developed to optimize the detection of COVID-19 pneumonitis from CXRs.

Details of this step-wise training approach include that in the first stage of pretraining, a custom CNN and selected ImageNet-pretrained CNN models are retrained on a large selection of CXRs with sufficient diversity due to sourcing from different collections, to coarsely learn the characteristics of normal and abnormal lungs. This CXR-specific pretraining helps in converting the weight layers, specific to the CXRs, in subsequent steps. The motivation behind

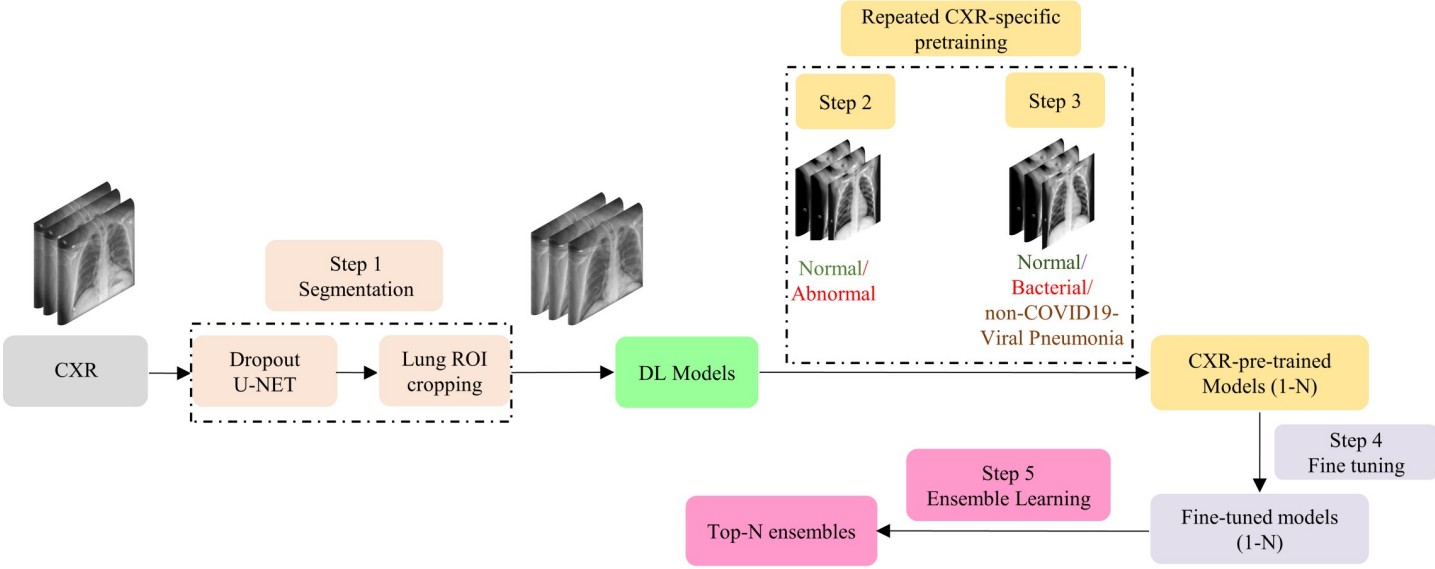

**Fig 3. The workflow of the proposed repeated CXR-specific pretraining and fine-tuning.**

this approach is to perform a knowledge transfer from the natural image domain to CXR-domain and learn the characteristics of normal lungs and a wide selection of CXR-specific pulmonary disease manifestations. During this training step, the datasets are split at the patient-level into 90% for training and 10% for testing. We randomly allocated 10% of the training data for validation.

During the second stage of repeated CXR-specific pretraining, the learned knowledge from the first stage pretrained models is transferred and repurposed to classify CXRs as exhibiting normal lungs, bacterial pneumonia, or non-COVID-19 viral pneumonia manifestations. This pretraining is motivated by the biological similarity in non-COVID-19 viral and COVID-19 viral pneumonia. However, there exist distinct radiological manifestations between each other as well as with non-viral pneumonia-related opacities [6, 29]. The motivation is to transfer the learned knowledge and fine-tune for COVID-19 detection. For the normal class, we pooled CXRs from various collections to introduce generalization and improve model performance. During this pretraining stage, again, the datasets are split at the patient-level into 90% for training and 10% for testing. For validation, we randomly allocated 10% of the training data.

The learned knowledge from the second stage of pretraining is transferred and fine-tuned to improve performance in classifying CXRs as showing normal lungs or COVID-19 viral pneumonia disease manifestations. Table 2 shows the datasets and their distribution used in various stages of learning proposed in this study. We compare this performance to that without repeated CXR-specific pretraining, referred to as *Baseline*. In the Baseline data set the ImageNet-pretrained CNNs are retrained out-of-the-box to categorize the CXRs as showing normal lungs or COVID-19 viral disease manifestations. For the normal class, we pooled CXRs in a patient-specific manner from various collections to introduce generalization and improve model performance. During this training step, we performed a patient-level split of the train and test data as follows: The CXRs from the Montreal-COVID-19 and Twitter-COVID-19 collections are combined ($n = 360$) where $n$ is the total number of images in the collection. The data are split at the patient-level into 80% for training and 20% for testing. We randomly allocated 10% of the training data for validation. The test set includes 72 CXRs, containing 36 CXRs each from the Montreal-COVID-19 and Twitter-COVID-19 collections. The

**Table 2. Datasets and their distribution used in various stages of learning.**

| Dataset | Normal | Abnormal | Bacterial pneumonia | Non-COVID-19 viral pneumonia | COVID-19+ |
|---|---|---|---|---|---|
| First stage of repeated CXR-specific pretraining | | | | | |
| RSNA | 8331 | 17833 | - | - | - |
| CheXpert | 16480 | 17000 | - | - | - |
| NIH | 59892 | 51708 | - | - | - |
| Total | 84703 | 86541 | - | - | - |
| Second stage of repeated CXR-specific pretraining | | | | | |
| RSNA | 400 | - | - | - | - |
| CheXpert | 400 | - | - | - | - |
| NIH | 400 | - | - | - | - |
| Pediatric CXR | 1583 | - | 2780 | 1493 | - |
| Total | 2783 | - | 2780 | 1493 | - |
| COVID-19 detection | | | | | |
| RSNA | 120 | - | - | - | - |
| CheXpert | 120 | - | - | - | - |
| NIH | 120 | - | - | - | - |
| Montreal-COVID-19 | - | - | - | - | 226 |
| Twitter-COVID-19 | - | - | - | - | 134 |
| Total | 360 | - | - | - | 360 |

In the first stage of repeated CXR-specific pretraining, a custom CNN and a selection of ImageNet-pretrained CNNs are retrained on a large selection of CXRs to learn CXR-specific features to categorize them as showing normal or abnormal lungs. During the second stage of repeated CXR-specific pretraining, the first-stage pretrained models are retrained on a collection of CXRs to categorize them as showing normal lungs, bacterial pneumonia, or non-COVID-19 viral pneumonia manifestations. Note that the pediatric CXR dataset predates the onset of the SARS-CoV2 virus, and therefore the viral pneumonia is of non-COVID-19 type. During the COVID-19 detection stage, the second-stage pretrained models are fine-tuned to classify CXRs into showing normal lungs or COVID-19 viral patterns.

GT disease annotations for this test data are set by the verification of publicly identified cases from two expert radiologists, referred to as Rad-1 and Rad-2 hereafter, with a combined experience of 60 years. The radiologists used the web-based VGG Image Annotator tool [36] to independently annotate the test collection by manually setting boundary boxes for what they believed to be COVID-19-related abnormalities. This was done in independent sessions in which each radiologist was shown the chest radiographs in Portable Network Graphics format with a spatial resolution of 1024 × 1024 pixels and was asked to annotate COVID-19 viral disease-specific ROI in the given test set.

It is well known that large amounts of high-quality data are imperative for DL model training and achieving superior performance. A challenge in the medical image-based DL is the lack of sufficient data. Many studies limit their work to data sourced from a single site. Using limited, single-site data toward model training may result in loss of generalizability and degrade model performance when evaluated on unseen data from other institutions or diverse imaging practices. Under these circumstances, generalizability and performance could be improved by increasing the variability of training data. In this study, we use a diversified data distribution from multiple CXR collections to enhance model generalization and performance in repeated CXR-specific pretraining and fine-tuning stages. Class weights are used to reward the minority classes to prevent biasing error and reduce overfitting. During model training, data are augmented with random horizontal and vertical pixel shifts in the range (-5 to 5) and rotations in the degree range (-9 to 9).

The following CNN-based DL models were trained and evaluated at various stages of learning performed in this study: (i) a custom wide residual network (WRN) [37] with dropout, (ii)

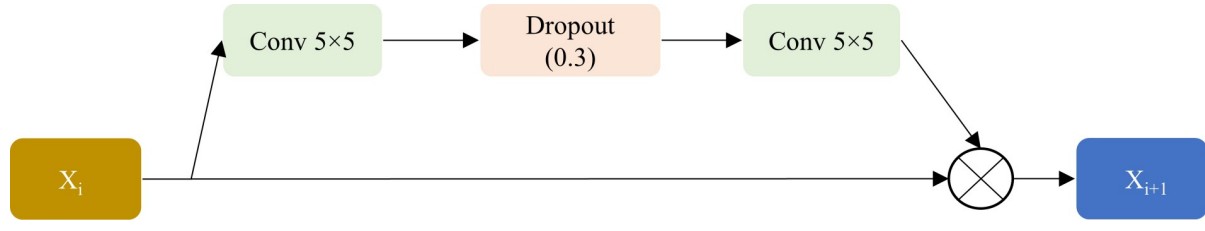

**Fig 4. A custom wide residual network (WRN) with dropout regularization.**

ResNet-18 [38], (iii) VGG-16 [39], (iv) VGG-19 [39], (v) Xception [40], (vi) Inception-V3 [41], (vii) DenseNet-121 [42], (viii) MobileNet-V2 [43], (ix) NasNet-Mobile [44]. The models are selected with an idea of increasing the architectural diversity, thereby increasing the representation power, when used in ensemble learning. All computation is done on a Windows® system with Intel Xeon CPU E3-1275 v6 3.80 GHz processor and NVIDIA GeForce® GTX 1050 Ti. We used Keras DL framework with Tensorflow backend, CUDA, and CUDNN libraries to accelerate GPU performance.

Residual CNNs having depths of hundreds of layers suffer from diminishing feature reuse [37]. This occurs due to issues with gradient flow, which results in only a few residual blocks learning useful feature representations. A WRN combats diminishing feature reuse issues by reducing the number of layers and increasing model width [37]. The resultant networks are found to exhibit shorter training times with similar or improved accuracy. In this study, we use a custom WRN with dropout regularization. Dropouts provide restrictive regularization, address overfitting issues, and enhance generalization. After empirical observations, we used $5 \times 5$ kernels for the convolutional layers, assigned a dropout ratio of 0.3, a depth of 16, and a width of 4, for the custom WRN used in this study. Fig 4 shows a WRN block with the dropout used in this study. The output from the deepest residual block is average pooled, flattened, and appended to a final dense layer with Softmax activation to predict class probabilities.

As mentioned before, ImageNet-pretrained CNNs have been developed for computer vision tasks using natural images. These models have varying depth and learn diversified feature representations. For medical images that are often available in limited quantities, deeper models may not be optimal and can lead to overfitting and loss of generalization. During the first stage of pretraining, the CNNs are instantiated with their ImageNet-pretrained weights and are truncated at empirically determined intermediate layers to effectively learn the underlying feature representations for CXR images and improve classification performance. The truncated models are appended with (i) zero-padding, (i) a $3 \times 3$ convolutional layer with 1024 feature maps, (ii) a global average pooling (GAP) layer, (iii) a dropout layer with an empirically determined dropout ratio of 0.5, and (iv) a final dense layer with Softmax activation to output prediction probabilities. These customized models learn CXR-specific feature representations

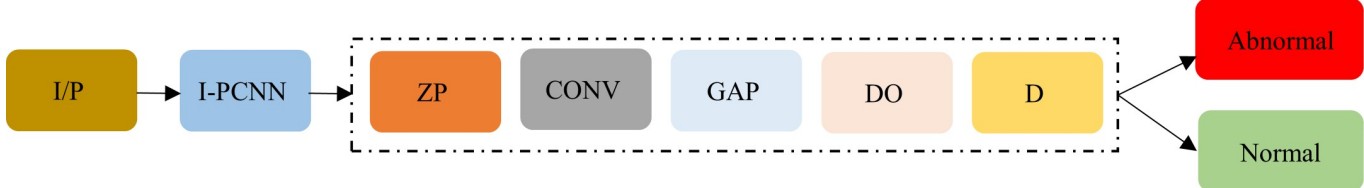

**Fig 5. The architecture of the CNNs used in the first stage of repeated CXR-specific pretraining.** I/P = Input, I-PCNN = truncated ImageNet-pretrained CNNs, ZP = Zero-padding, CONV = Extra convolution layer, GAP = Global Average Pooling, DO = Dropout, D = Final dense layer with Softmax activation.

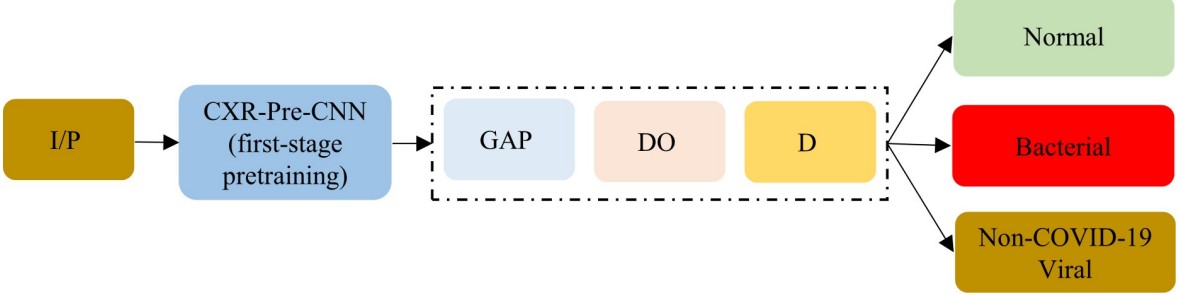

**Fig 6. The architecture of the CNNs used in the second stage of pretraining.** I/P = Input, CXR-Pre-CNN = CXR-specific CNNs from the first stage of pretraining, truncated at their deepest convolutional layer, GAP = Global Average Pooling, DO = Dropout, D = Final dense layer with Softmax activation.

to classify CXR images as showing normal or abnormal lungs. The custom WRN is initialized with random weights. Fig 5 shows the architecture of the pretrained CNNs used during the first stage of repeated CXR-specific pretraining.

In the second stage, pretrained models from the first stage are truncated at their deepest convolutional layer and appended with (i) GAP layer, (ii) dropout layer (ratio = 0.5), and (iii) dense layer with Softmax activation to output class probabilities for CXRs showing normal lungs, bacterial pneumonia, or non-COVID-19 viral pneumonia. Fig 6 shows the architecture of the customized models used during the second stage of pretraining.

Next, the second-stage pretrained models are truncated at their deepest convolutional layer and appended with (i) GAP layer, (ii) dropout layer (ratio = 0.5), and (iii) dense layer with Softmax activation. The resultant models are fine-tuned to classify the CXRs as belonging to COVID-19+ or normal classes where '+' symbolizes COVID-19-positive cases. Fig 7 shows the architecture of the models used toward COVID-19 detection.

The models in various learning stages are trained and evaluated using stochastic gradient descent (SGD) optimization to estimate learning error and classification performance. We used callbacks to check the internal states of the models and store model checkpoints. The model weights delivering superior performance with the test data are used for further analysis. The performance of the models at various learning stages is evaluated using the following metrics: (i) Accuracy; (ii) Area under curve (AUC); (iii) Sensitivity; (iv) Specificity; (v) Precision; (vi) $F_1$ score; (vii) Matthews correlation coefficient (MCC); (viii) Kappa statistic; and (ix) Diagnostic Odds Ratio (DOR). The following ensemble strategies are applied to the fine-tuned models for COVID-19 detection to improve performance: (i) Majority voting; (ii) Simple averaging; and (iii) Weighted averaging. In majority voting, the predictions with maximum votes are considered as final predictions. The average of the individual model predictions is considered the final prediction in a simple averaging ensemble. For a weighted ensemble, we

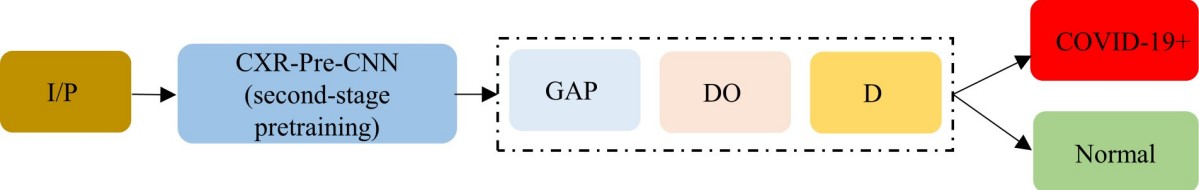

**Fig 7. The architecture of the CNNs fine-tuned toward COVID-19 detection.** I/P = Input, CXR-Pre-CNN = CXR-pretrained CNNs from the second stage of pretraining, truncated at their deepest convolutional layer, GAP = Global Average Pooling, DO = Dropout, D = Final dense layer with Softmax activation.

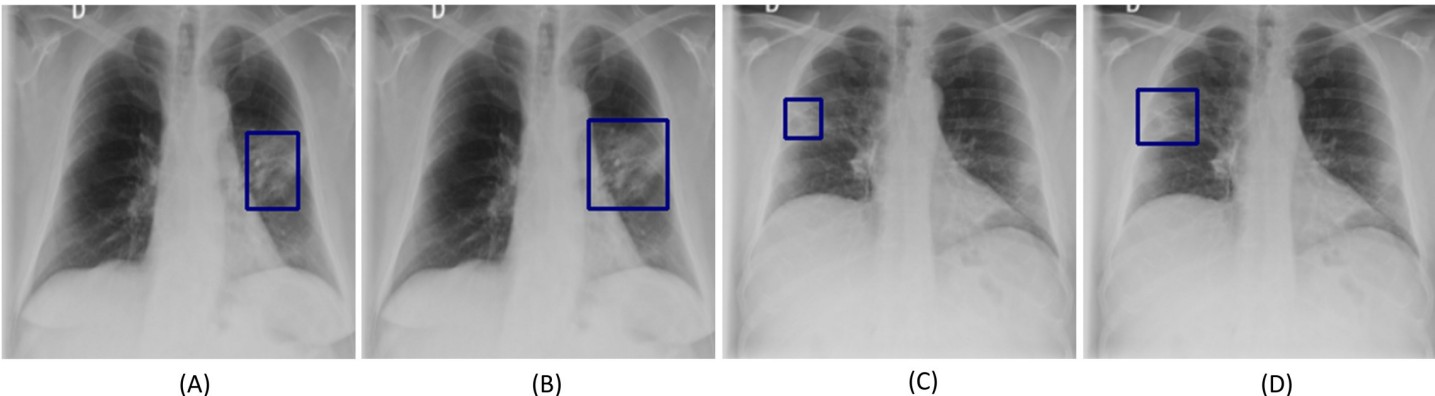

**Fig 8. Examples showing inter-reader variability in annotating COVID-19 disease ROI.** (A) and (B) show the annotations (bounding boxes in blue) of Rad-1 and Rad-2, respectively, for a given COVID-19 disease labeled image; (C) and (D) shows the GT annotations of Rad-1 and Rad-2, respectively for another COVID-19 disease labeled image.

optimized the weights for the model predictions that minimized the total logarithmic loss. This loss decreases as the prediction probabilities converge to GT labels. We used the Sequential Least Squares Programming (SLSQP) algorithmic method [45] to perform several iterations of constrained logarithmic loss minimization to converge to the optimal weights for the model predictions.

## Inter-reader variability analysis

Fig 8 shows examples of COVID-19 viral disease-specific ROI annotations on CXRs made by Rad-1 and Rad-2. In this study, we used the well-known Simultaneous Truth and Performance Level Estimation (STAPLE) algorithm [13] to arrive at a consensus reference ROI annotation and use it to evaluate the performance of the top-N ensembles and to simultaneously assess the performance against each radiologist.

STAPLE methods are widely used in validating image segmentation algorithms and comparing the performance of experts. Segmentation solutions are treated as a response to a pixel-wise classification problem. The algorithm uses an expectation-maximization (EM) approach

**Table 3. Performance metrics achieved during the first-stage of CXR-specific pretraining.**

| Models | Acc. | AUC (CI) | Sens. | Spec. | Prec. | $F_1$ | MCC | Kappa | DOR |
|---|---|---|---|---|---|---|---|---|---|
| Custom WRN | 0.6696 | 0.722 (0.7153, 0.7287) | 0.6566 | 0.6828 | 0.6763 | 0.6663 | 0.3395 | 0.3393 | 4.12 |
| VGG-16 | 0.6874 | 0.7397 (0.7331, 0.7463) | 0.6641 | 0.711 | 0.6988 | 0.6810 | 0.3755 | 0.3750 | 4.87 |
| VGG-19 | **0.6913** | **0.7435 (0.7374, 0.7506)** | 0.6651 | 0.7178 | 0.704 | 0.6840 | **0.3833** | **0.3827** | **5.06** |
| Inception-V3 | 0.6842 | 0.7375 (0.7309, 0.7441) | 0.6186 | 0.7506 | 0.7145 | 0.6631 | 0.3723 | 0.3689 | 4.89 |
| Xception | 0.6727 | 0.7287 (0.7220, 0.7354) | 0.6364 | 0.7094 | 0.6885 | 0.6614 | 0.3466 | 0.3456 | 4.28 |
| DenseNet-121 | 0.6827 | 0.7416 (0.7350, 0.7482) | **0.7589** | 0.606 | 0.6603 | **0.7062** | 0.3692 | 0.3650 | 4.85 |
| NasNet-Mobile | 0.6820 | 0.7347 (0.7281, 0.7413) | 0.5802 | **0.7849** | **0.7313** | 0.6471 | 0.3728 | 0.3647 | 5.05 |
| MobileNet-V2 | 0.6844 | 0.7426 (0.7360, 0.7492) | 0.7007 | 0.668 | 0.6805 | 0.6904 | 0.3688 | 0.3686 | 4.72 |
| ResNet-18 | 0.6821 | 0.7338 (0.7272, 0.7404) | 0.7307 | 0.6332 | 0.6679 | 0.6979 | 0.3657 | 0.3640 | 4.69 |

The custom WRN is initialized with random weights. Data in parenthesis are 95% CI for the AUC values measured as the exact Clopper–Pearson interval corresponding to separate 2-sided CI with individual coverage probabilities of $\sqrt{0.95}$. (Acc. = Accuracy, AUC = Area under curve, Sens. = Sensitivity, Spec. = Specificity, Prec. = Precision, $F_1$ = $F_1$ score, MCC = Matthews correlation coefficient, DOR = Diagnostics odd ratio). Bold numerical values denote best performances in the respective columns. None of these individual differences are statistically significant.

that computes a probabilistic estimate of a reference segmented image computed from a collection of expert annotations and weighing them by an estimated level of performance for each expert. It incorporates this knowledge to spatially distribute the segmented structures while satisfying homogeneity constraints. The details pertaining to the algorithm and the performance measures including Kappa statistic, sensitivity, specificity, positive predictive value (PPV), and negative predictive value (NPV) used to analyze inter-reader variability and assess program performance are summarized in Section A of the S1 File.

### Disease ROI localization

In this study, we use CRM [15] visualization to evaluate the effectiveness of CRM-based ensemble localization. Details of the CRM algorithm are provided in Section B of the S1 File. First, we use CRM-based ROI localization to interpret predictions of individual CNNs and compare against the GT annotations provided by each expert. Next, we select the top-3, top-5, and top-7 performing models, construct ensemble CRMs through an averaging process and compare against each radiologists' independent annotations, and the STAPLE-generated consensus annotation. Finally, we quantitatively compare the ensemble localization performance with each other and against individual CRMs in terms of IoU and mean average precision (mAP) metrics. The mAP score is calculated by taking the mean of average precision (AP) over various IoU thresholds [46].

### Statistical analysis

Statistical tests were conducted to determine significance in performance differences between the models. We used confidence intervals (CI) to measure model discrimination capability and estimate its precision through the error margin. We measured 95% CI as the exact Clopper–Pearson interval for the AUC values obtained by the models in various learning stages. Statistical packages including StatsModels and SciPy are used in these analyses. We performed a one-way analysis of variance (ANOVA) [47] on mAP values obtained with the top-N (N = (3, 5, 7)) model ensembles to study their localization performance and determine statistical significance among them and against the annotations of each of the radiologist and also the STAPLE-generated consensus ROI annotation. One-way ANOVA tests are performed only if the assumptions of data normality and homogeneity of variances are satisfied for which we performed Shapiro-Wilk and Levene's analyses [47]. Statistical analyses are performed using R statistical software (Version 3.6.1).

### Results

Recall that in the first stage of CXR-specific pretraining, we truncated the ImageNet-pretrained CNNs at their intermediate layers to empirically determine the layers that demonstrated superior performance. These empirically determined layers for the various models are listed in Section C of the S1 File. The performance achieved through truncating the models at the selected intermediate layers and appending task-specific heads toward classifying the CXRs is shown in Table 3.

From Table 3, we observe that the AUC values are not statistically significantly different across the models ($p > 0.05$). The DOR provides a measure of diagnostic accuracy and estimation of discriminative power. A high DOR is obtained by a model that exhibits high sensitivity and specificity with low FPs and FNs. A model with higher AUC indicates that it is more capable of distinguishing TNs and TPs. Considering DOR and AUC values, VGG-19 demonstrates somewhat better performance followed by NasNet-Mobile in classifying CXRs into normal or abnormal categories. Also considering MCC and Kappa metrics, VGG-19 outperformed other

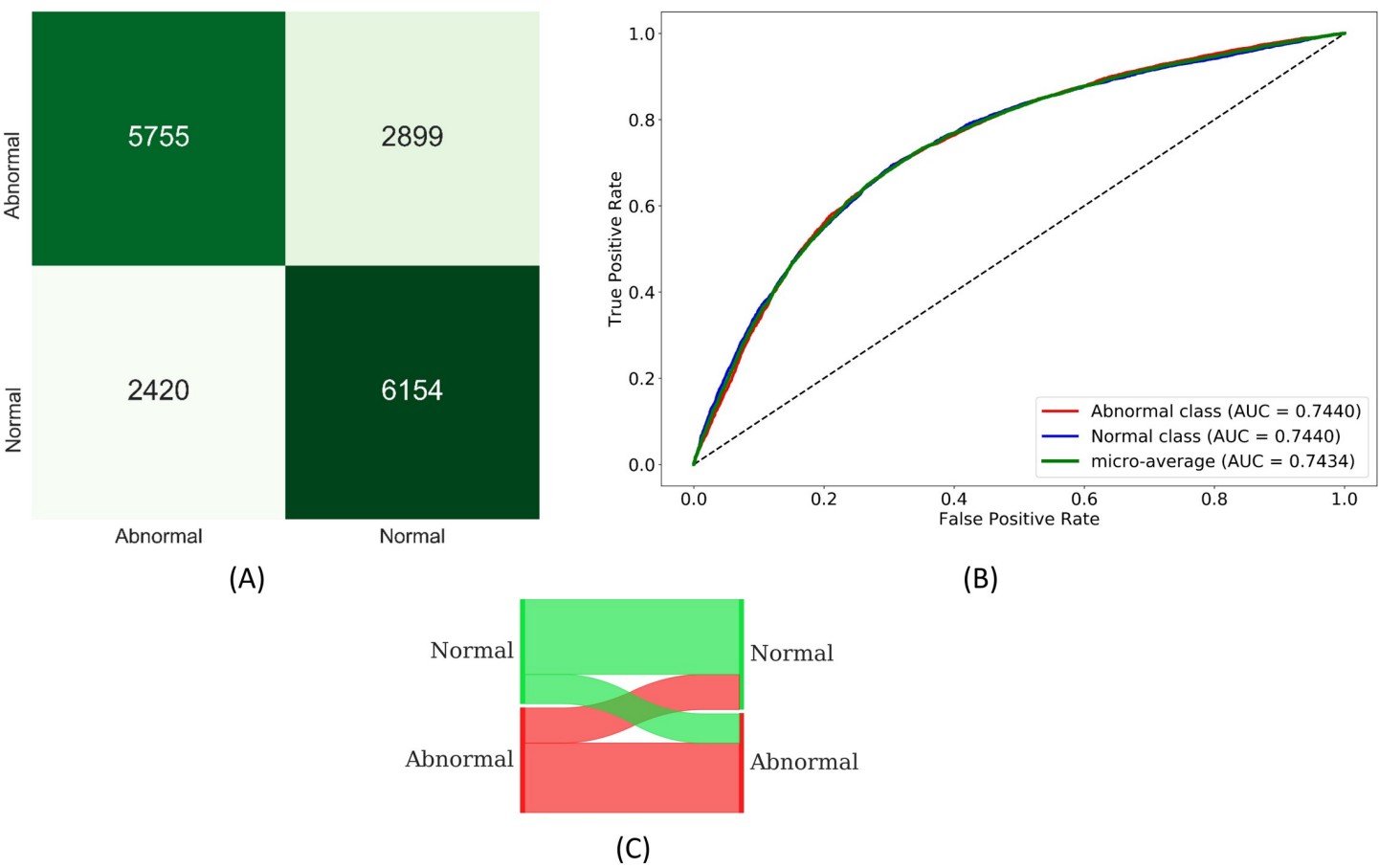

**Fig 9. Performance achieved using the VGG-19 model during the first-stage of CXR-specific pretraining.** (A) Confusion matrix; (B) ROC curves; (C) Normalized Sankey flow diagram.

models. The confusion matrix, ROC curves, and normalized Sankey flow diagram obtained using the VGG-19 model toward this classification task are shown in Fig 9. We used a normalized Sankey diagram [48] to visualize model performance. Here, weights are assigned to the classes on the truth (left) and prediction (right) side of the diagram to provide an equal visual

**Table 4. Performance metrics achieved by the models during the second stage of CXR-specific pretraining.**

| Models | Acc. | AUC (CI) | Sens. | Spec. | Prec. | $F_1$ | MCC | Kappa | DOR |
|---|---|---|---|---|---|---|---|---|---|
| Custom WRN | 0.7007 | 0.8589 (0.8332, 0.8846) | 0.7007 | 0.8068 | 0.74 | 0.671 | 0.5326 | 0.5136 | 9.78 |
| VGG-16 | 0.8879 | 0.9735 (0.9616, 0.9854) | 0.8879 | 0.9298 | 0.896 | 0.8773 | 0.8312 | 0.8214 | 104.91 |
| VGG-19 | 0.8922 | 0.9739 (0.9621, 0.9857) | 0.8922 | 0.9304 | 0.906 | 0.8825 | 0.8389 | 0.8281 | 110.64 |
| Inception-V3 | 0.9135 | 0.9792 (0.9699, 0.9895) | 0.9135 | 0.9518 | 0.9120 | 0.9110 | 0.8656 | 0.8644 | 180.97 |
| Xception | 0.905 | 0.9714 (0.9590, 0.9838) | 0.905 | 0.943 | 0.9064 | 0.9017 | 0.8532 | 0.8503 | 157.61 |
| DenseNet-121 | **0.9177** | **0.9835 (0.9740, 0.9930)** | **0.9177** | **0.9519** | **0.9187** | **0.9141** | **0.8736** | **0.8704** | **220.68** |
| NasNet-Mobile | 0.9163 | 0.9819 (0.9720, 0.9918) | 0.9163 | 0.9477 | 0.9222 | 0.9106 | 0.8674 | 0.8674 | 198.38 |
| MobileNet-V2 | 0.9121 | 0.9812 (0.9711, 0.9913) | 0.9121 | 0.952 | 0.9113 | 0.9098 | 0.8637 | 0.8621 | 205.81 |
| ResNet-18 | 0.8936 | 0.9738 (0.9620, 0.9856) | 0.8936 | 0.9329 | 0.8997 | 0.8849 | 0.8383 | 0.8309 | 116.77 |

Bold numerical values denote best performances in the respective columns. None of these individual differences are statistically significant.

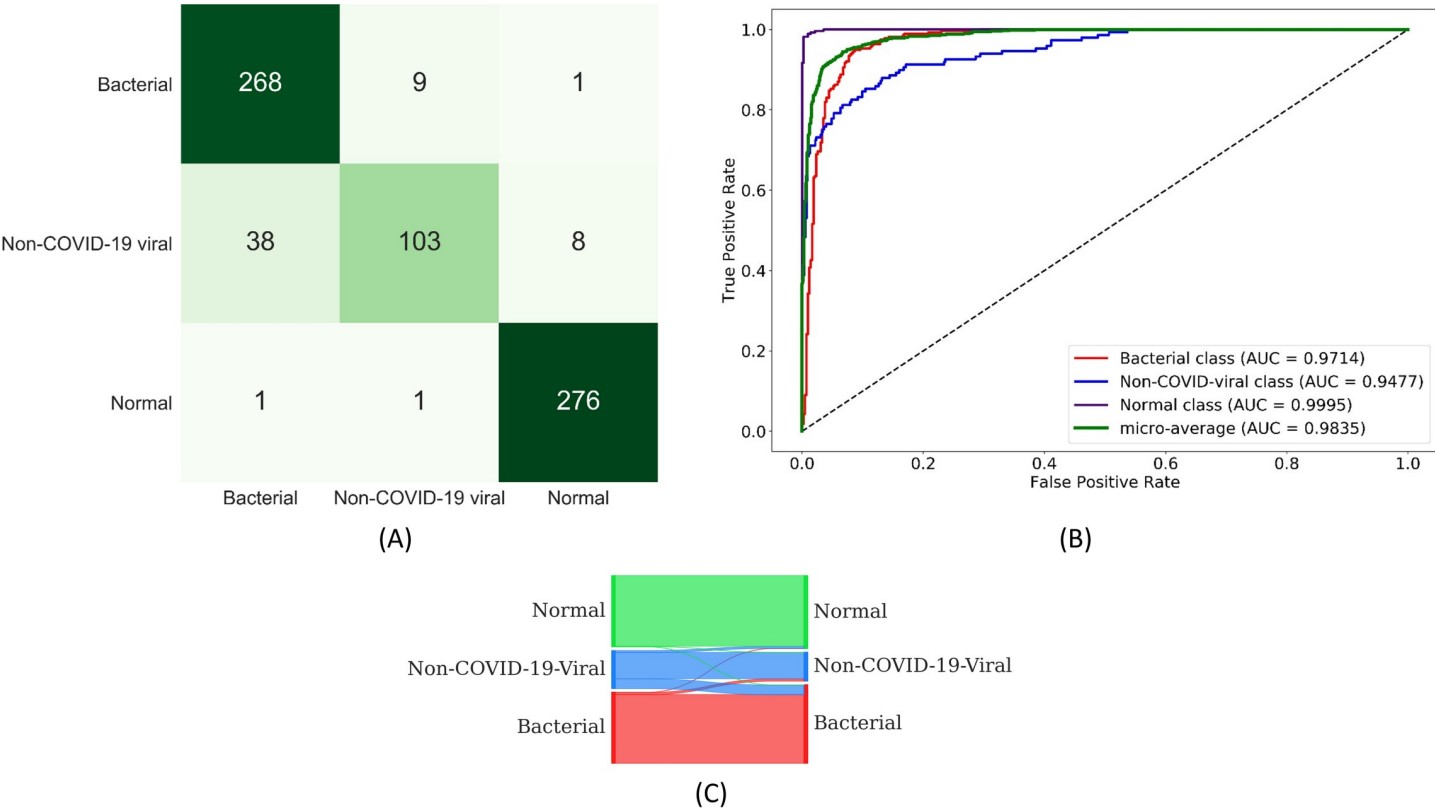

**Fig 10. Performance achieved using the DenseNet-121 model during the second stage of CXR-specific pretraining.** (A) Confusion matrix; (B) ROC curves; (C) Normalized Sankey flow diagram.

representation for the classes on either side. The strips width changes across the plot so that the width of each at the right side represents the fraction of all objects which the model predicts as belonging to a category that truly belongs to each of the categories.

Recall that during the second stage of CXR-specific pretraining, the learned representations from the first-stage pretrained models are transferred and fine-tuned to classify CXRs as

**Table 5. Performance metrics achieved with fine-tuning the second-stage pretrained models for COVID-19 detection.**

| Models | Acc. | AUC (CI) | Sens. | Spec. | Prec. | $F_1$ | MCC | Kappa | DOR |
|---|---|---|---|---|---|---|---|---|---|
| D-WRN | 0.8333 | 0.9043 (0.8562, 0.9524) | **0.9028** | 0.7639 | 0.7927 | 0.8442 | 0.6732 | 0.6667 | 30.06 |
| VGG-16 | 0.8681 | 0.9302 (0.8885, 0.9719) | 0.8473 | 0.8889 | 0.8841 | 0.8653 | 0.7368 | 0.7361 | 44.4 |
| VGG-19 | 0.8611 | 0.9176 (0.8726, 0.9626) | 0.9028 | 0.8195 | 0.8334 | 0.8667 | 0.7248 | 0.7222 | 42.17 |
| Inception-V3 | 0.8611 | 0.9123 (0.8660, 0.9586) | **0.9028** | 0.8195 | 0.8334 | 0.8667 | 0.7248 | 0.7222 | 42.17 |
| Xception | 0.8681 | 0.9297 (0.8879, 0.9715) | 0.8334 | 0.9028 | 0.8956 | 0.8634 | 0.7379 | 0.7361 | 46.47 |
| DenseNet-121 | 0.875 | 0.9386 (0.8993, 0.9779) | **0.9028** | 0.8473 | 0.8553 | 0.8784 | 0.7512 | 0.75 | 51.54 |
| NasNet-Mobile | 0.8542 | 0.911 (0.8644, 0.9576) | 0.8612 | 0.8473 | 0.8494 | 0.8552 | 0.7085 | 0.7083 | 34.43 |
| MobileNet-V2 | 0.875 | 0.925 (0.8819, 0.9681) | 0.8473 | 0.9028 | 0.8971 | 0.8715 | 0.7512 | 0.75 | 51.54 |
| ResNet-18 | **0.8958** | **0.9490 (0.9132, 0.9854)** | 0.8612 | **0.9306** | **0.9254** | **0.8921** | **0.7936** | **0.7917** | **83.2** |

Bold numerical values denote best performances in the respective columns. Overall, ResNet-18 showed the best performance but individual metrics are not statistically different from other models.

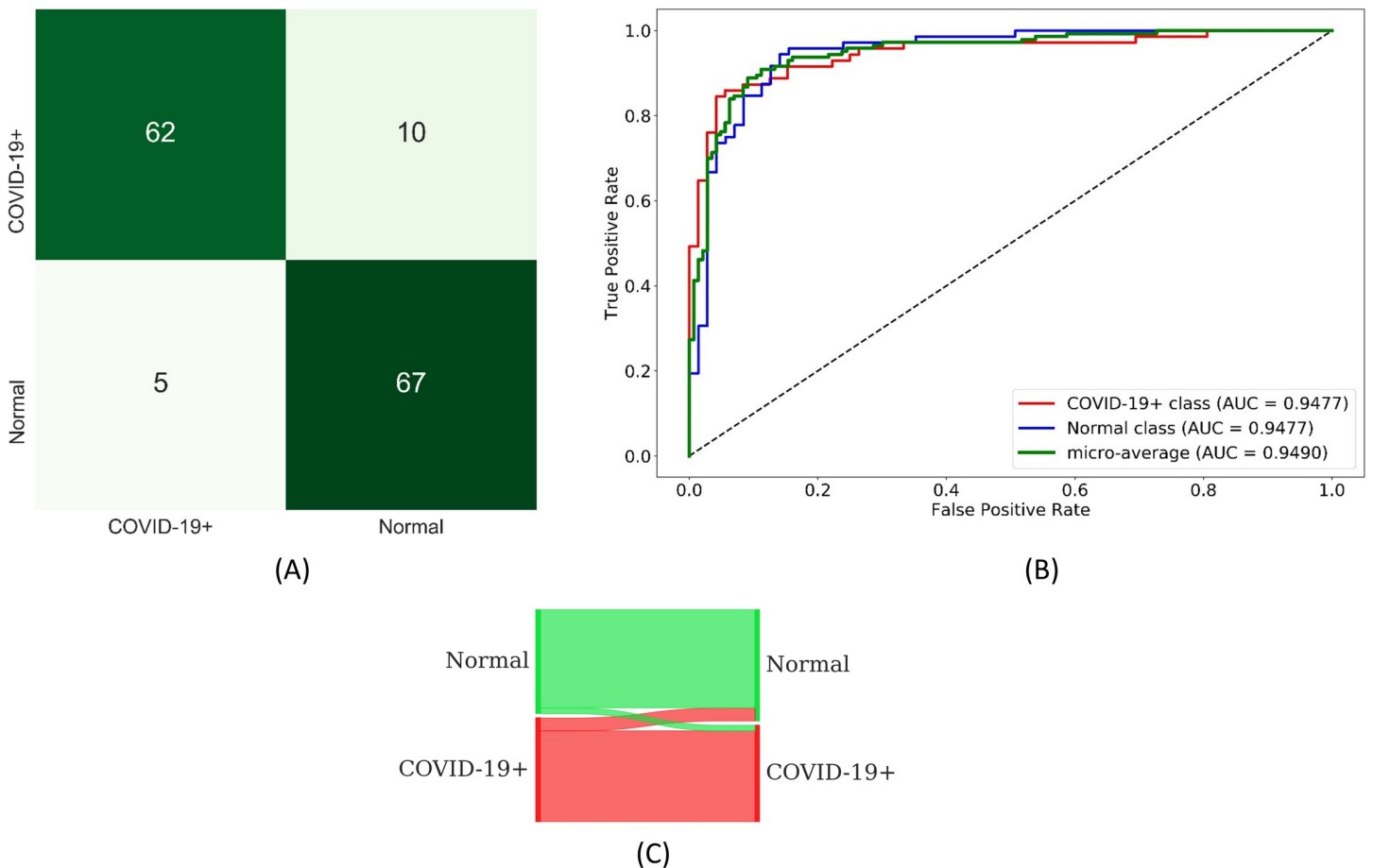

**Fig 11. Performance achieved using the ResNet-18 model during fine-tuning for COVID-19 detection.** (A) Confusion matrix; (B) ROC curves; (C) Normalized Sankey flow diagram.

showing normal lungs, bacterial proven pneumonia, or non-COVID-19 viral pneumonia. The performance achieved by the second-stage pretrained models is shown in Table 4.

We observed no statistically significant difference in AUC values achieved with the models during this pretraining stage ($p > 0.05$). Considering DOR, DenseNet-121 demonstrated better performance (220.68) followed by MobileNet-V2 (205.81) in categorizing the CXRs as showing normal lungs, bacterial pneumonia, or non-COVID-19 viral pneumonia. Considering MCC and $F_1$ score metrics that consider both sensitivity and precision to determine model generalization, DenseNet-121 outperformed other models. The confusion matrix, ROC curves, and normalized Sankey flow diagram obtained using the DenseNet-121 model toward this classification task are shown in Fig 10.

The second stage pretrained models are truncated at their deepest convolutional layer, appended with task-specific heads, and fine-tuned to classify the CXRs as belonging to COVID-19+ or normal categories. Table 5 shows the performance metrics achieved by the models toward this task.

We observed no statistically significant difference in AUC values ($p > 0.05$) achieved by the fine-tuned models. Considering DOR, ResNet-18 demonstrated better performance (83.2) followed by DenseNet-121 (51.54) in categorizing the CXRs as showing normal lungs or manifesting COVID-19 viral disease. The custom WRN, Inception-V3, and DenseNet-121 are

**Table 6. Performance metrics achieved during fine-tuning the second-stage pretrained models for COVID-19 detection is compared with the baseline.**

| Models | Method | Acc. | AUC (CI) | Sens. | Spec. | Prec. | $F_1$ | MCC | Kappa | DOR | Para. Reduction (%) |
|---|---|---|---|---|---|---|---|---|---|---|---|
| Custom WRN | Baseline | 0.7897 | 0.8014 (0.7362, 0.8666) | 0.6742 | 0.8675 | 0.8396 | 0.7478 | 0.5611 | 0.5433 | 14.34 | - |
| | Fine-tuned | 0.8333 | **0.9043 (0.8562, 0.9524)** | 0.9028 | 0.7639 | 0.7927 | 0.8442 | 0.6732 | 0.6667 | 30.06 | 0 |
| VGG-16 | Baseline | 0.7708 | 0.7993 (0.7338, 0.8648) | 0.6667 | 0.875 | 0.8422 | 0.7442 | 0.5539 | 0.5416 | 14.01 | - |
| | Fine-tuned | 0.8681 | **0.9302 (0.8885, 0.9719)** | 0.8473 | 0.8889 | 0.8841 | 0.8653 | 0.7368 | 0.7361 | 44.4 | 0 |
| VGG-19 | Baseline | 0.7847 | 0.8176 (0.7545, 0.8807) | 0.8334 | 0.7362 | 0.7595 | 0.7948 | 0.5722 | 0.5694 | 13.97 | - |
| | Fine-tuned | 0.8611 | **0.9176 (0.8726, 0.9626)** | 0.9028 | 0.8195 | 0.8334 | 0.8667 | 0.7248 | 0.7222 | 42.17 | 0 |
| Inception-V3 | Baseline | 0.8472 | 0.9285 (0.8864, 0.9706) | 0.8473 | 0.8473 | 0.8473 | 0.8473 | 0.6945 | 0.6944 | 30.79 | - |
| | Fine-tuned | 0.8611 | 0.9123 (0.8660, 0.9586) | 0.9028 | 0.8195 | 0.8334 | 0.8667 | 0.7248 | 0.7222 | 42.17 | **42.36** |
| Xception | Baseline | 0.8472 | 0.9215 (0.8775, 0.9655) | 0.9028 | 0.7917 | 0.8125 | 0.8553 | 0.6988 | 0.6944 | 35.31 | - |
| | Fine-tuned | 0.8681 | 0.9297 (0.8879, 0.9715) | 0.8334 | 0.9028 | 0.8956 | 0.8634 | 0.7379 | 0.7361 | 46.47 | **37.57** |
| DenseNet-121 | Baseline | 0.8333 | 0.9153 (0.8698, 0.9608) | 0.9028 | 0.7639 | 0.7927 | 0.8442 | 0.6732 | 0.6667 | 30.06 | - |
| | Fine-tuned | 0.8750 | 0.9386 (0.8993, 0.9779) | 0.9028 | 0.8473 | 0.8553 | 0.8784 | 0.7512 | 0.75 | 51.54 | **54.51** |
| NasNet-Mobile | Baseline | 0.7778 | 0.8502 (0.7919, 0.9085) | 0.8473 | 0.7084 | 0.744 | 0.7923 | 0.561 | 0.5556 | 13.48 | - |
| | Fine-tuned | 0.8542 | **0.911 (0.8644, 0.9576)** | 0.8612 | 0.8473 | 0.8494 | 0.8552 | 0.7085 | 0.7083 | 34.43 | **11.85** |
| MobileNet-V2 | Baseline | 0.8681 | 0.9325 (0.8915, 0.9735) | 0.8473 | 0.8889 | 0.8841 | 0.8653 | 0.7368 | 0.7361 | 44.4 | - |
| | Fine-tuned | 0.8750 | 0.925 (0.8819, 0.9681) | 0.8473 | 0.9028 | 0.8971 | 0.8715 | 0.7512 | 0.75 | 51.54 | **37.38** |
| ResNet-18 | Baseline | 0.8542 | 0.9302 (0.8885, 0.9719) | 0.9167 | 0.7917 | 0.8149 | 0.8628 | 0.714 | 0.7083 | 41.83 | - |
| | Fine-tuned | 0.8958 | 0.9477 (0.9130, 0.9850) | 0.8612 | 0.9306 | 0.9254 | 0.8921 | 0.7936 | 0.7917 | 83.2 | **46.05** |

The Baseline refers to retraining out-of-the-box ImageNet-pretrained CNNs toward this task. Bold numerical values show the models that achieved a significantly better AUC compared to baseline and the models that showed a reduction in the number of parameters.

found to be equally sensitive (0.9028) toward this classification task. However, the ResNet-18 fine-tuned model demonstrated better performance with other performance metrics including accuracy, AUC, specificity, precision, $F_1$ score, MCC, and Kappa. The confusion matrix, ROC

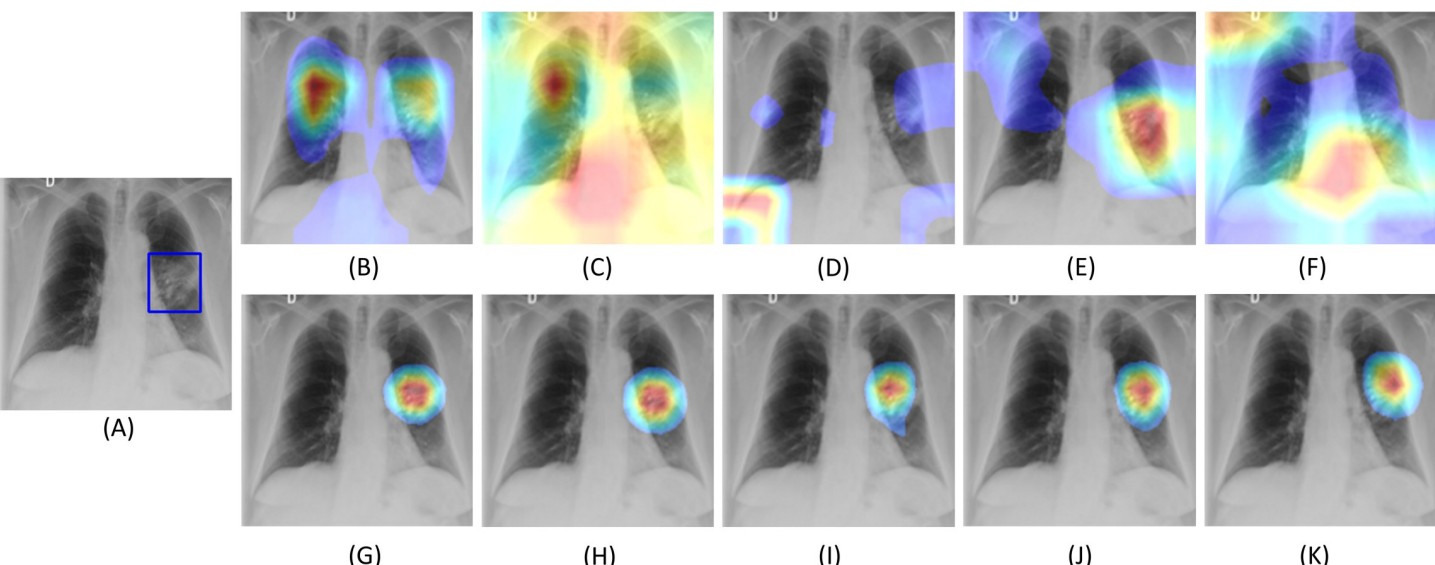

**Fig 12. COVID-19 viral disease ROI CRM-based localization achieved using the fine-tuned models and their baseline counterparts.** (A) Original CXR with STAPLE-generated consensus ROI (shown as blue box ROI); (B) Baseline VGG-16; (C) Baseline VGG-19; (D) Baseline MobileNet-V2; (E) Baseline ResNet-18; (F) Baseline Inception-V3; (G) Fine-tuned VGG-16; (H) Fine-tuned VGG-19; (I) Fine-tuned MobileNet-V2; (J) Fine-tuned ResNet-18; (K) Fine-tuned Inception-V3.

**Table 7. Performance achieved with an ensemble of top-3, top-5, and top-7 fine-tuned models toward COVID-19 detection.**

| Ensemble method | Top-N models | Acc. | AUC (CI) | Sens. | Spec. | Prec. | F₁ | MCC | Kappa | DOR |
|---|---|---|---|---|---|---|---|---|---|---|
| Majority voting | 3 | 0.9028 | 0.9097 (0.8628, 0.9566) | 0.8612 | 0.9167 | 0.9155 | 0.8986 | 0.8084 | 0.8055 | 102.22 |
| | 5 | 0.8819 | 0.8819 (0.8291, 0.9347) | 0.8612 | 0.9028 | 0.8986 | 0.8795 | 0.7646 | 0.7639 | 57.63 |
| | 7 | 0.8889 | 0.8889 (0.8375, 0.9403) | 0.875 | 0.9028 | 0.9000 | 0.8874 | 0.7781 | 0.7778 | 65.02 |
| Simple averaging | 3 | 0.8958 | 0.9483 (0.9121, 0.9845) | 0.8889 | 0.9028 | 0.9015 | 0.8952 | 0.7918 | 0.7917 | 74.32 |
| | 5 | 0.8819 | 0.9462 (0.9093, 0.9831) | 0.8612 | 0.9028 | 0.8986 | 0.8795 | 0.7646 | 0.7639 | 57.63 |
| | 7 | 0.8819 | 0.9453 (0.9081, 0.9825) | 0.875 | 0.8889 | 0.8874 | 0.8812 | 0.764 | 0.7639 | 56.01 |
| Weighted averaging | **3** | **0.9097** | **0.9508 (0.9118, 0.9844)** | **0.9028** | **0.9445** | **0.9394** | **0.9091** | **0.8196** | **0.8194** | **105.6** |
| | 5 | 0.9028 | 0.9493 (0.9134, 0.9852) | 0.875 | 0.9306 | 0.9265 | 0.9000 | 0.8069 | 0.8055 | 93.87 |
| | 7 | 0.8889 | 0.9459 (0.9089, 0.9829) | 0.8889 | 0.8889 | 0.8889 | 0.8889 | 0.7778 | 0.7778 | 64.02 |

Bold numerical values denote best performances in the respective columns. Top-3 weighted averaging looks best but the AUC differences are not statistically significant.

curves, and normalized Sankey flow diagram obtained using the ResNet-18 model toward this classification task are shown in Fig 11.

We visualized the deepest convolutional layer feature embedding for the ResNet-18 fine-tuned model, using the t-Distributed Stochastic Neighbor Embedding (t-SNE) algorithm [49], which is shown in Section D of the S1 File. The performance obtained with the fine-tuned models is compared to the *Baseline*, as shown in Table 6. The Baseline refers to out-of-the-box ImageNet-pretrained CNNs that are retrained toward this classification task. The custom WRN is initialized with randomized weights for the Baseline task.

As observed in Table 6, the fine-tuned models achieved better performance compared to their baseline counterparts. The AUC metrics achieved with the fine-tuned custom WRN, VGG-16, VGG-19, and NasNet-Mobile models are shown in bold type and are observed to be statistically better than ($p < 0.05$) their baseline, untuned counterparts. We also observed a marked reduction in the number of trainable parameters for the fine-tuned models. The fine-tuned DenseNet-121 model showed a 54.51% reduction in the number of trainable parameters while delivering better performance as compared to its baseline counterpart. The same holds true for ResNet-18 (46.05%), Inception-V3 (42.36%), Xception (37.57%), MobileNet-V2 (37.38%), and NasNet-Mobile (11.85%) with the added benefit of improved performance compared to their baseline models.

We performed visualization studies to compare how the fine-tuned models and their baseline counterparts localize the ROIs in a CXR manifesting COVID-19 viral patterns. Fig 12 shows the following: (i) a CXR with COVID-19 disease consensus ROI obtained with STAPLE using Rad-1 and Rad-2 annotations, and (ii) the ROI localization achieved with various fine-tuned models and their baseline counterparts.

We extracted the features from the deepest convolution layer of the fine-tuned models and their baseline counterparts. We used CRM tools to localize the pixels involved in predicting the CXR images as showing COVID-19 viral disease patterns. As observed in Fig 12, the baseline models demonstrated poor disease ROI localization, compared to the fine-tuned models. We observed that the fine-tuned models learned salient ROI feature representations, matching the experts' knowledge about the disease ROI. The localization excellence of the fine-tuned models can be attributed to (i) CXR-specific knowledge transfer that helped to learn modality-specific characteristics; the learned feature representations are transferred and repurposed for the COVID-19 detection task, and (ii) optimal architecture depth to learn the salient ROI feature representations to classify CXRs to their respective categories. These deductions are supported by poor localization performance of deeper, out-of-the-box ImageNet-pretrained

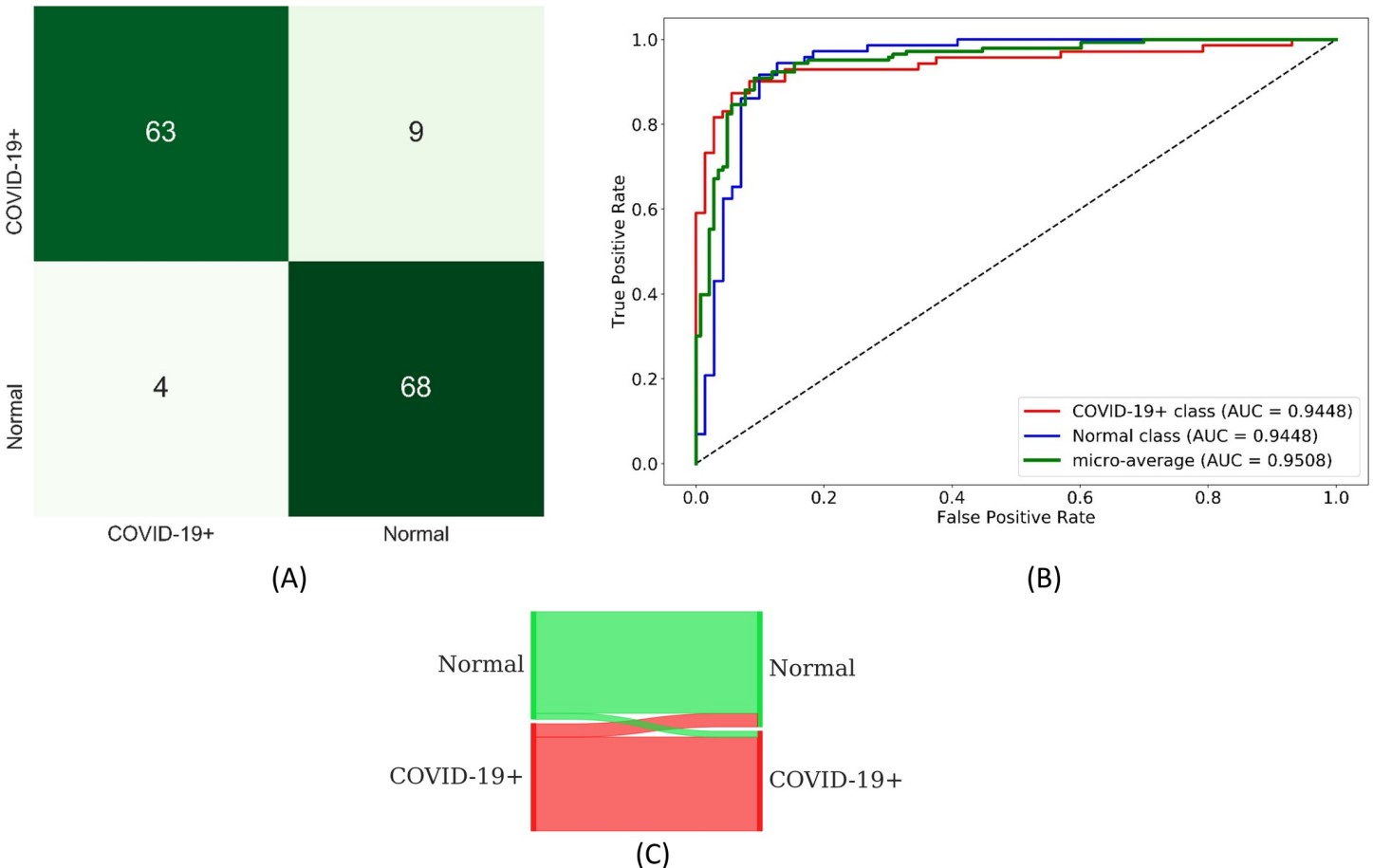

**Fig 13. Performance achieved through weighted averaging of the top-3 fine-tuned CNNs toward COVID-19 detection.** (A) Confusion matrix; (B) ROC curves; (C) Normalized Sankey flow diagram.

baseline CNNs like ResNet-18, Inception-V3, and MobileNet-V2, which possibly suffered from baseline overfitting that resulted in poor learning and generalization.

**Table 8. Performance achieved in terms of CRM-based IoU and mAP values by the individual fine-tuned CNNs using the radiologists' annotations and STAPLE-generated ROI consensus annotation.**

| Annotations | Parameters | Xception | Inception-V3 | DenseNet-121 | VGG-19 | VGG-16 | MobileNet-V2 | ResNet-18 | NasNet-Mobile |
|---|---|---|---|---|---|---|---|---|---|
| Rad-1 | IOU | 0.0678 | **0.1174** | 0.0799 | 0.0854 | 0.1076 | 0.0644 | 0.0972 | 0.1000 |
| | mAP@[0.1:0.7] | 0.0571 | **0.1142** | 0.0697 | 0.0645 | 0.0986 | 0.0712 | 0.0593 | 0.075 |
| | Ranking | 8 | **1** | 5 | 6 | 2 | 4 | 7 | 3 |
| Rad-2 | IOU | 0.2146 | 0.2567 | 0.2398 | 0.2183 | 0.2230 | 0.1825 | 0.2293 | **0.2569** |
| | mAP@[0.1:0.7] | 0.146 | 0.206 | 0.1858 | 0.1643 | 0.1882 | 0.1467 | 0.1742 | **0.2186** |
| | Ranking | 8 | 2 | 4 | 6 | 3 | 7 | 5 | **1** |
| STAPLE | IOU | 0.0670 | **0.1337** | 0.0916 | 0.0951 | 0.1267 | 0.0713 | 0.1126 | 0.1095 |
| | mAP@[0.1:0.7] | 0.0603 | **0.1213** | 0.0792 | 0.073 | 0.1068 | 0.0775 | 0.0648 | 0.0851 |
| | Ranking | 8 | **1** | 4 | 6 | 2 | 5 | 7 | 3 |

Bold numerical values denote best performances in the respective rows.

**Table 9. IOU and mAP values obtained with top-3, top-5, and top-7 ensembles using annotations of Rad-1, Rad-2, and STAPLE-generated consensus ROI annotations.**

| Annotations | Parameters | Top-3 | Top-5 | Top-7 |
|---|---|---|---|---|
| Rad-1 | IOU | **0.1343** | 0.0994 | 0.1236 |
| | mAP@[0.1:0.7] | **0.1264** | 0.0767 | 0.0753 |
| Rad-2 | IOU | 0.2673 | **0.2955** | 0.2865 |
| | mAP@[0.1:0.7] | 0.2179 | **0.2352** | 0.2292 |
| STAPLE | IOU | **0.1518** | 0.1193 | 0.1350 |
| | mAP@[0.1:0.7] | **0.1352** | 0.0924 | 0.0916 |

Bold numerical values denote best performances in the respective rows.

We constructed ensembles of the top-3, top-5, and top-7 performing fine-tuned CNNs to evaluate for an improvement in predicting the CXRs as showing normal lungs or COVID-19 viral disease patterns. We used majority voting, simple averaging, and weighted averaging strategies toward this task. In weighted averaging, we optimized the weights for the model predictions to minimize the total logarithmic loss. We used the SLSQP algorithm to iterate through this minimization process and converge to the optimal weights for the model predictions. The results achieved with the various ensemble methods are shown in Table 7. We observed no statistically significant difference in the AUC values achieved by the various ensemble methods ($p > 0.05$). We observed that the performance with top-3 ensembles is better than that of top-5 and top-7 ensembles. It is observed that the weighted averaging of top-3 fine-tuned CNNs viz. ResNet-18, MobileNet-V2, and DenseNet-121 demonstrated better performance when their predictions are optimally weighted at 0.6357, 0.1428, and 0.2216, respectively. This weighted averaging ensemble delivered better performance in terms of accuracy, AUC, DOR, Kappa, $F_1$ score, MCC, and other metrics, as compared to other ensembles. The confusion matrix, ROC curves, and normalized Sankey flow diagram obtained with the weighted averaging of the top-3 fine-tuned CNNs are shown in Fig 13.

Table 8 shows the performance achieved in terms of CRM-based IoU and mAP scores by the individual fine-tuned CNNs using the annotations of Rad-1, Rad-2, and STAPLE-generated consensus ROI. For Rad-1, the fine-tuned Inception-V3 model demonstrated higher values for the average IoU and mAP metrics. For Rad-2, we observed that the fine-tuned NasNet-Mobile outperformed other models. With STAPLE-generated consensus ROI, the Inception-V3 model outperformed other models in localizing COVID-19 viral disease-specific ROI.

The precision-recall (PR) curves of the best performing models using Rad-1, Rad-2, and the STAPLE-generated consensus ROI are shown in Section E of the S1 File. These curves are generated for varying IoU thresholds in the range (0.1–0.7). This range is empirically determined from the PR curves to alleviate issues due to poor and high sensitivity and precision rates and ensure measuring mAP scores to appropriately reflect the models' localization ability. The confidence score threshold is varied to generate each curve. For a given fine-tuned model, we define the confidence score as the highest heat map value in the predicted ROI weighted by the classification score at the output nodes. We considered the ROI predictions as TP when the IoU and confidence scores are higher than their corresponding thresholds. For a given PR curve, we computed the AP score as the average of the precision across all recall values.

The following are the important observations from this localization study: The accuracy of a model is not related to disease ROI localization. From Table 6, we observed that the fine-tuned ResNet-18 model is highly accurate, followed by DenseNet-121 and MobileNet-V2, in classifying the CXRs as belonging to the COVID-19 viral category. However, while localizing

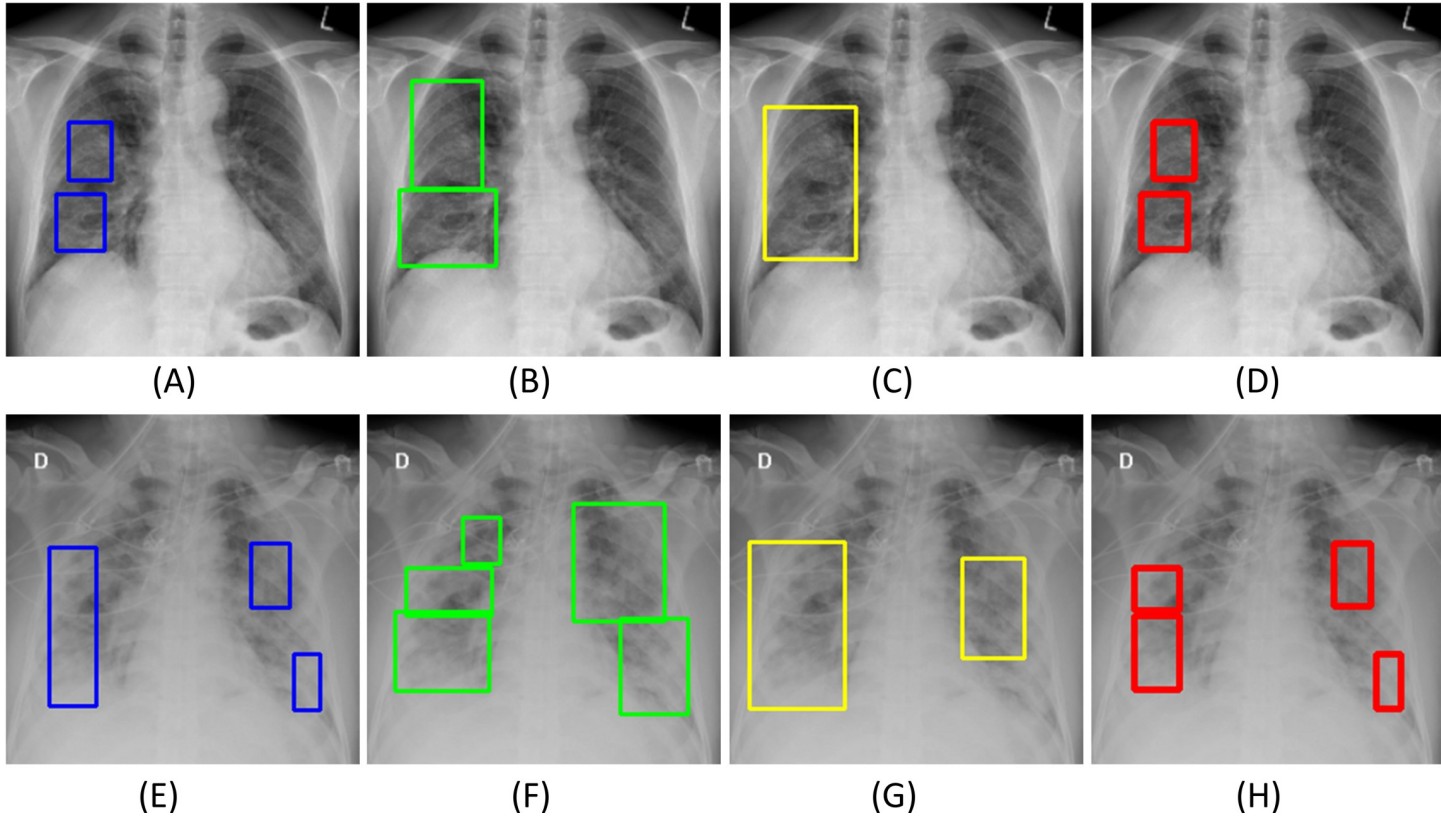

**Fig 14. Sample CXRs from two different patients (rows A-D and E-H, respectively) show ROI annotations generated.** (A) and (E) Rad-1 (in blue); (B) and (F) Rad-2 (in green); (C) and (G) Top-3 ensemble using STAPLE-generated consensus ROI (program) (in yellow); (D) and (H) STAPLE-generated consensus ROI annotation (in red).

disease-specific ROI, the Inception-V3, VGG-16, and NasNet-Mobile fine-tuned models delivered superior ROI localization performance compared to other models. This underscores the fact that the classification accuracy of a model is not an optimal measure to interpret its learned behavior. Localization studies are indispensable to understand the learned features and compare them to the expert knowledge for the problem under study. These studies provide comprehensive qualitative and quantitative measures of the learning capacity of the model and its generalization ability.

Next, we constructed an ensemble of CRMs through averaging the ROI localization by the top-3, top-5, and top-7 fine-tuned models. We ranked the models based on the IoU and mAP scores. The localization performance achieved with the various ensemble CRMs is shown in Table 9. We observed that the ensemble CRMs delivered superior ROI localization performance compared to that achieved with the individual models. However, the number of models in the top-performing ensembles varied. While using the annotations of Rad-1, we observed that the ensemble of the top-3 models demonstrated higher values for IoU and mAP than other ensembles. However, for Rad-2, the ensemble of the top-5 models demonstrated superior localization with IoU and mAP values of 0.2955 and 0.2352, respectively. The ensemble of top-3 fine-tuned models demonstrated higher values for IoU and mAP scores compared to other models while using STAPLE-generated ROI consensus annotation. Considering this study, we observed that averaging the CRMs of more than top-5 fine-tuned models didn't

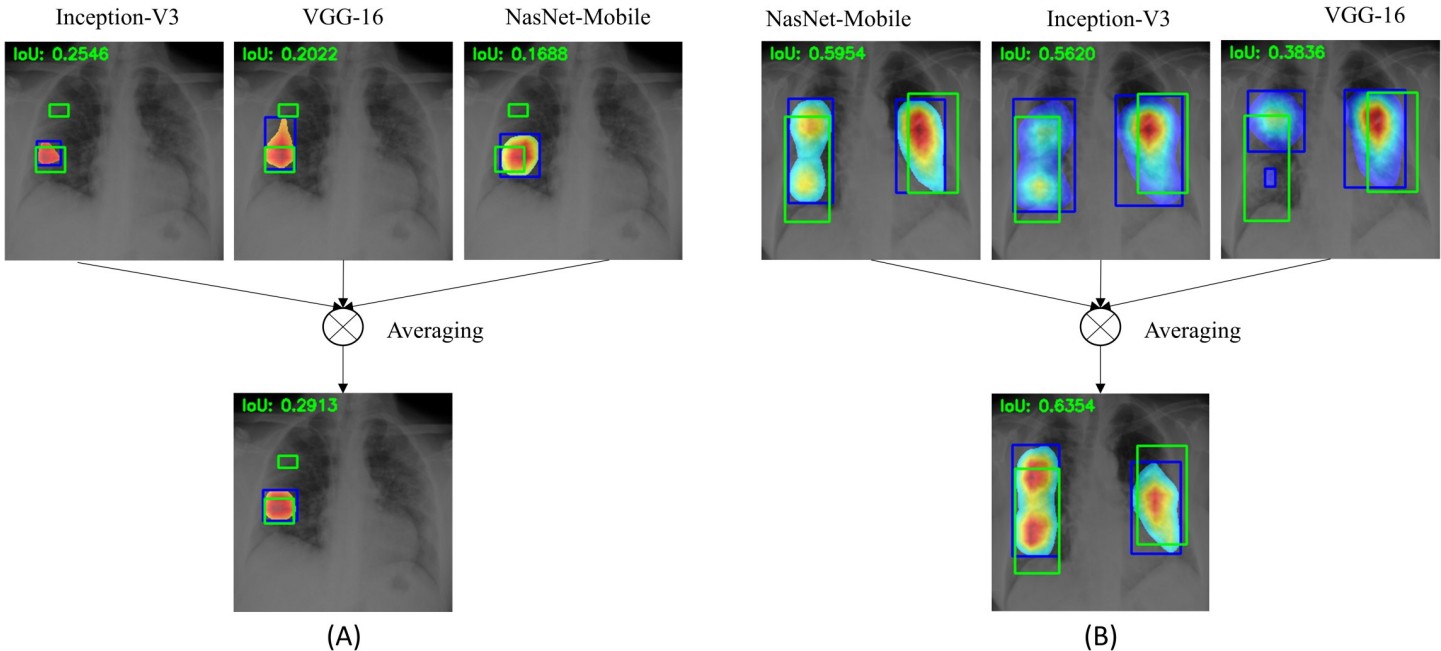

**Fig 15. Instances of ensemble CRMs combining top-N ensemble ROI predictions.** (A) top-3 CNNs using STAPLE-generated consensus ROI annotation; (B) top-5 CNNs using Rad-2 annotations. The green box denotes reference ROI annotation and the blue box denotes ensemble CRM localization.

improve performance but rather it saturates ROI localization. PR curves resulting from this observation are shown in Section F of the S1 File.

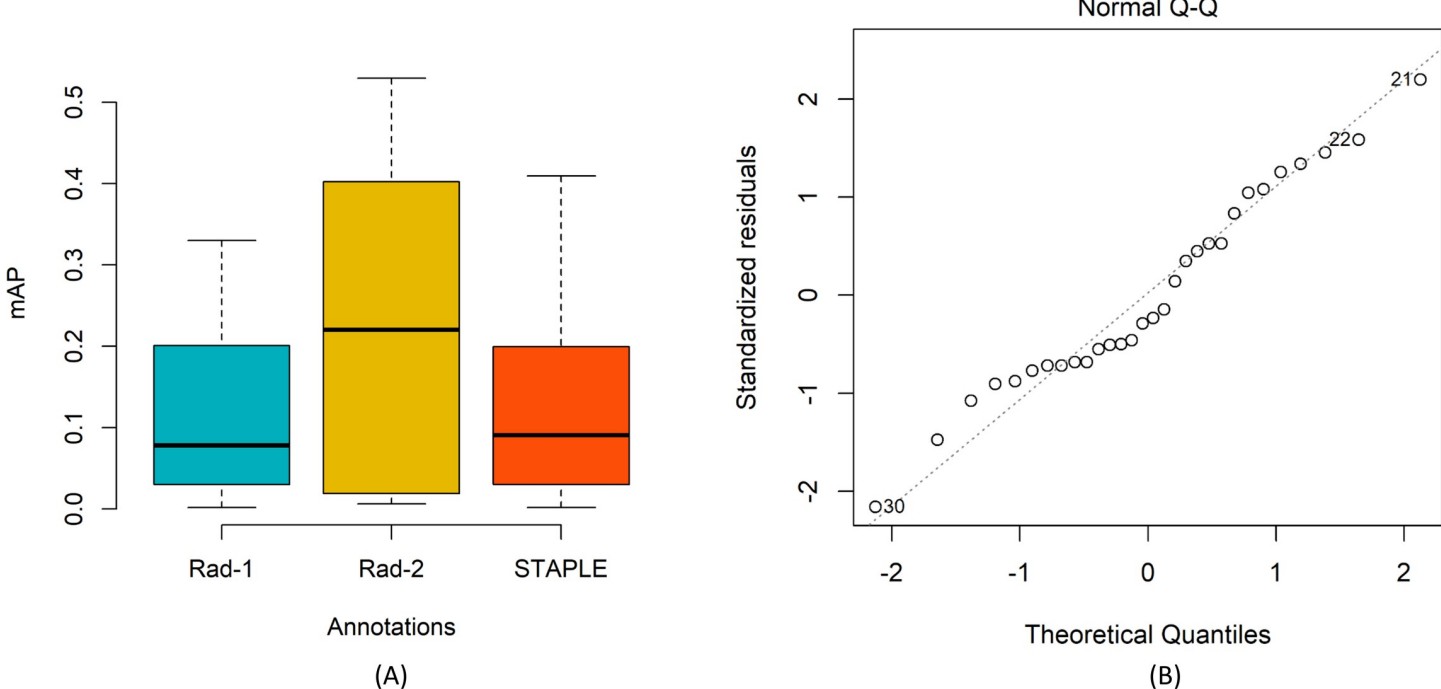

**Fig 16. Statistical analyses.** (A) Mean plot for the mAP scores obtained by the top-N ensembles using Rad-1, Rad-2, and STAPLE-generated consensus ROI annotations; Error bars represent standard errors. The differences are not statistically significant; (B) Residual plot showing the data follow the normal distribution.

**Table 10. Consolidated results of Shapiro–Wilk, Levene, and one-way ANOVA analyses.**

| Metric | Shapiro–Wilk ($p$) | Levene's test ($p$) | ANOVA (F) | ANOVA ($p$) |
|--------|--------------------|---------------------|-----------|-------------|
| mAP | 0.1014 | 0.3365 | 1.678 | 0.2060 |

Instances of CXRs showing ROI annotations of Rad-1, Rad-2, top-3 ensemble using STA-PLE-generated ROI consensus (referred to as *program* hereafter), and the STAPLE-generated ROI consensus annotation are shown in Fig 14.

Fig 15 shows the following: (A) an ensemble CRM generated with the top-3 fine-tuned models that delivered superior localization performance using STAPLE-generated ROI consensus annotation, and (B) an ensemble CRM generated with the top-5 fine-tuned models that delivered superior localization performance using the annotations of Rad-2.

We observe that the CRMs obtained using individual models in the top-N ensemble highlight ROI to varying extents. The ensemble CRM averages the ROIs localized with individual CRMs to highlight the disease-specific ROI involved in class prediction. The ensemble CRMs have a superior IoU value, compared to that of individual CRMs; the ensemble CRM improved localization performance as compared to individual ROI localization. This underscores the fact that ensemble localization improves performance and ability to generalize, conforming to the experts' knowledge about COVID-19 viral disease manifestations.

To perform a one-way ANOVA analysis, we investigated whether the assumptions of data normality and homogeneous variances are satisfied. We used the Shapiro–Wilk test to investigate for normal distribution of the data and Levene's test, for homogeneity of variances, using mAP scores obtained with the top-N ensembles. We plotted the residuals to investigate if the assumption of normal residual distribution is satisfied. Fig 16 shows the following: (A) The mean plot for the mAP scores obtained by the top-N ensembles using Rad-1, Rad-2, and STA-PLE-generated consensus ROI annotations, and (B) a plot of the quantiles of the residuals against that of the normal distribution.

It is observed from the residual plot shown in Fig 16 that all the points fall approximately along with a 45-degree reference. This underscores the fact that the assumption of normal distribution of data is satisfied. Table 10 shows the consolidated results of Shapiro–Wilk, Levene, and one-way ANOVA analyses.

To compute one-way ANOVA, we measure the variance between group means, the variance within the group, and the group sizes. This information is combined to measure statistical significance from the test statistic F. In our study, we have three groups (Rad-1, Rad-2, and STAPLE) of 10 observations each, hence the distribution is mentioned as F (2, 27). As observed from Table 10, the $p$-values obtained with the Shapiro-Wilk test are not significant ($p > 0.05$) and reveal that the normality assumption is satisfied. The result of Levene's test is not statistically significant ($p > 0.05$). This demonstrates that the variance across the mAP values obtained with the annotations of Rad-1, Rad-2, and STAPLE-generated consensus ROI are not statistically significantly different. Since the conditions of data normality and homogeneity of variances are satisfied, we performed one-way ANOVA to explore the existence of a statistically significant difference in the mAP scores. To this end, we observed no statistically significant difference in the mAP scores obtained with Rad-1, Rad-2, and STAPLE-generated consensus ROI (F (2, 27) = 1.678, $p = 0.2060$). This smaller F-value underscores the fact that the null hypothesis ($H_0$), i.e., that all groups demonstrate equal mAP scores, holds good.

We used the STAPLE-generated consensus ROI as to the standard reference and measured its agreement with that generated by the program and the radiologists. The consensus ROI is estimated from the set of ROI annotations provided by Rad-1 and Rad-2. STAPLE assumes

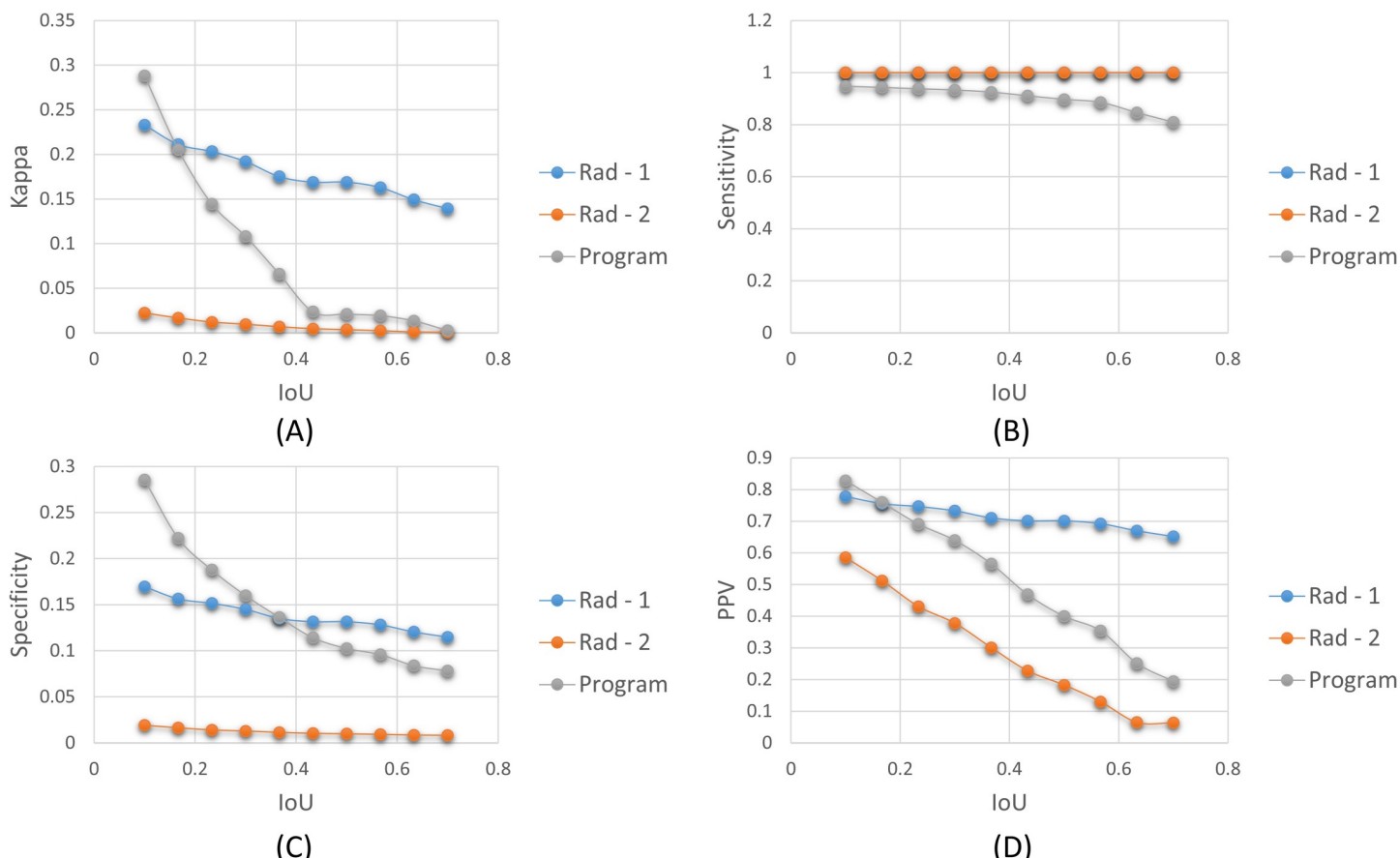

**Fig 17. Assessing inter-reader variability and program performance.** The following performance metrics are measured and plotted for 10 different IoU thresholds in the range (0.1–0.7): (A) Kappa statistic; (B) Sensitivity; (C) Specificity; (D) PPV.

that Rad-1 and Rad-2 individually annotated ROIs for the given CXRs so that the quality of annotations are captured. We determined the set of TPs, FPs, TNs, and FNs for 10 different IoU thresholds in the range (0.1–0.7) and provided a measure of inter-reader variability and program performance using the following metrics: (i) Kappa statistic; (ii) Sensitivity; (iii) Specificity; (iv) PPV; and (v) NPV. These parameters depend on the relative proportion of the disease-specific ROI. An ROI provided by a radiologist or predicted by the program is considered as a TP if the IoU with the consensus ROI is greater than or equal to a given IoU threshold. Each radiologist or program ROI that produces an IoU less than the threshold or falls outside the consensus ROIs is counted as FP. The FN is defined as a radiologist or program ROI that is completely missing when there is a consensus ROI. If there is an image with no ROIs on both

**Table 11. Performance level assessment and inter-reader variability analysis using STAPLE-generated consensus ROI.**

| Annotations | Kappa | Sensitivity | Specificity | PPV | NPV |
|---|---|---|---|---|---|
| Rad—1 | **0.1805** | **1.0** | 0.1384 | **0.7140** | **1.0** |
| Rad—2 | 0.0080 | **1.0** | 0.0121 | 0.2877 | **1.0** |
| Program | 0.0740 | 0.9037 | **0.1467** | 0.5154 | 0.6 |

Bold numerical values denote the best performances in respective columns.

the ROI annotations under test, it is considered as TN. Fig 17 shows the variability in Kappa, sensitivity, specificity, and PPV values observed for the Rad-1, Rad-2, and the program.

The estimated Kappa, sensitivity, specificity, PPV, and NPV values that are averaged over 10 different IoU thresholds in the range (0.1–0.7) are shown in Table 11.

The performance assessment as observed from Table 11 indicated that Rad-1 is more specific than Rad-2. The same holds good for the Kappa and PPV metrics. We observed that NPV is 1 for Rad-1 and Rad-2. This is because the number of FNs = 0, which signifies that none of the radiologists ROI completely missed when there is an ROI in the STAPLE-generated consensus annotation. However, the NPV achieved with the program is 0.6 which underscores the fact the predicted ROIs missed a marked proportion of ROIs in the STAPLE-generated consensus. This assessment indicated that Rad-1 generated annotations similar to that of STAPLE-generated consensus by demonstrating higher values for Kappa, sensitivity, and PPV as compared to Rad-2. We also observed that the program is performing with higher specificity but with lower sensitivity as compared to Rad-1 and Rad-2. These assessments provided feedback indicating the need for program modifications, parameter tuning, and other measures, to improve its localization performance.

## Discussion

There are several salient observations to be made from the analyses reported above. These include (i) the kind of data used in training, (ii) the size and variety of data collections, (iii) learning ability of various DL architectures informing their selection, (iv) need for customizing the models for improved performance, (v) benefits of ensemble learning, and (vi) the imperative need for localization to measure conformity to the problem.

We observed that repeated CXR-specific pretraining and fine-tuning resulted in improved performance toward COVID-19 detection as compared to the baseline, out-of-the-box, ImageNet pretrained CNNs. This highlights the need to use task-specific modality training resulting in improved model adaption, convergence, reduced bias, and reduced overfitting. This approach may have helped the DL models differentiate distinct radiological manifestations between COVID viral pneumonia and other non-viral pneumonia-related opacities. An added benefit is that this approach resulted in reductions in both computations and the number of trainable parameters.

It is well-known that neural networks develop or learn implicit rules to convert input data into features for making decisions. These learned rules are opaque to the user and the decisions are difficult to interpret. However, an interpretable model explaining its predictions related to model accuracy doesn't necessarily guarantee those accurate predictions are for the right reasons. Localization studies help observe if the model has learned salient ROI feature representations that agree with expert annotations. In our study, we demonstrate that CRM visualization tools show superior localization performance in localizing COVID-19 viral disease-specific ROIs, particularly for the fine-tuned models compared to the ImageNet-pretrained CNNs.

Model ensembles further improved qualitative and quantitative performance in COVID-19 detection. Ensemble learning compensated mislabeling in individual models by combining their predictions and reduced prediction variance to the training data. We observed that the weighted averaging ensemble of the top-3 performing fine-tuned models delivered better performance compared to any individual constituent model. The results demonstrate that the detection task benefits from an ensemble of repeated CXR-specific pretrained and fine-tuned models. Ensemble learning also compensates for localization errors in CRMs and missed ROIs by combining and averaging the individual CRMs. Empirical evaluations show that ensemble

localization demonstrated superior IoU and mAP scores and they significantly outperform ROI localization by individual CNN models.

It is difficult to quantify individual radiologists' performance in annotating ROIs in medical images. Not only are they the truth standard, but this "truth" is impacted by inherent biases related to a pandemic event like COVID-19 and their clinical exposure and experience. This complexity is compounded further because CXRs offer lower diagnostic sensitivity than CTs for example. So, a conservative assessment of the CXR is likely to result in smaller and more specific truth annotation ROIs. We used STAPLE to compute a probabilistic estimate of expert ROI annotations for the two expert radiologists who contributed to this study. STAPLE assumes these annotations are conditionally independent. The algorithm discovers and quantifies the bias among the experts when they differ in their opinion of the disease-specific ROI annotation. We use STAPLE-generated annotations as GT to assess the variation for every annotation for each expert, where the DL model is also considered as an expert. We observed that the Kappa values obtained using the STAPLE-generated consensus ROI are in a low range (0–0.2). This is probably because of the small number of experts and their inherent biases in assessing COVID-19 cases. Particularly, we note that Rad-1 was very specific in marking the ROIs, whereas Rad-2 annotated larger regions that sometimes accommodated multiple smaller regions into a single ROI. This led to lower IoU value that in turn affected the Kappa value. The pandemic is an evolving situation and CXR manifestations often exhibit biological similarity to non-COVID-19 viral pneumonia. The CXR is not a definitive diagnostic tool and expert views may differ in referring a candidate patient for further review. It would be helpful to conduct a similar analysis with a larger number of experts on a larger patient population. We remain hopeful that health agencies and medical societies will make such image collections available for future research. As more reliable and widely available COVID testing becomes available, the results of that testing could be used with CXRs as an additional important indicator of GT.

Regarding the limitations of our study: (i) The publicly available COVID-19 data collections used are fairly small and may not encompass a wide range of disease pattern variability. An appropriately annotated large-scale collection of CXRs with COVID-19 viral disease manifestations is necessary to build confidence in the models, improve their robustness, and generalization. (ii) The study is evaluated with the ROI annotations obtained from two expert radiologists. However, it would help to have more radiologists contribute independently in the annotation process and then arrive at a consensus that could reduce annotation errors. (iii) We used conventional convolutional kernels toward this study, however, future research could propose novel convolutional kernels that reduce feature dimensionality and redundancy and result in improved performance with reduced memory and computational requirements. (iv) Ensemble models require markedly high training time, memory, and computational resources for successful deployment and use. However, recent advancements in storage and computing solutions and cloud technology could lead to improvements in this regard.

## Conclusions

In this study, we have demonstrated that a combination of repeated CXR-specific pretraining, fine-tuning, and ensemble learning helped in (a) transferring CXR-specific learned knowledge that can be subsequently fine-tuned to improve COVID-19 detection in CXRs; and (b) improving classification generalization and localization performance by reducing prediction variance. Ensemble-based ROI localization helped in improving localization performance by compensating for the errors in individual constituent models. We also performed inter-reader variability analysis and program performance assessment by comparing them with a STAPLE-

based estimated reference. This assessment highlighted the opportunity for improving performance through ensemble modifications, requisite parameter optimization, increased task-specific dataset size, and involving "truth" estimates from a larger number of expert collaborators. We believe that the results proposed are useful for developing robust models for tasks involving medical image classification and disease-specific ROI localization.

## Supporting information

**S1 File. Supplementary material.**
(DOCX)

## Author Contributions

**Conceptualization:** Sivaramakrishnan Rajaraman, Sameer K. Antani.

**Data curation:** Philip O. Alderson, Les R. Folio, Sameer K. Antani.

**Formal analysis:** Sivaramakrishnan Rajaraman, Sameer K. Antani.

**Funding acquisition:** Sameer K. Antani.

**Investigation:** Sivaramakrishnan Rajaraman, Sudhir Sornapudi, Sameer K. Antani.

**Methodology:** Sivaramakrishnan Rajaraman, Sudhir Sornapudi, Sameer K. Antani.

**Project administration:** Sivaramakrishnan Rajaraman, Sameer K. Antani.

**Resources:** Sameer K. Antani.

**Software:** Sivaramakrishnan Rajaraman, Sudhir Sornapudi.

**Supervision:** Philip O. Alderson, Les R. Folio, Sameer K. Antani.

**Validation:** Sivaramakrishnan Rajaraman.

**Visualization:** Sivaramakrishnan Rajaraman, Sudhir Sornapudi.

**Writing – original draft:** Sivaramakrishnan Rajaraman, Sudhir Sornapudi.

**Writing – review & editing:** Sivaramakrishnan Rajaraman, Philip O. Alderson, Les R. Folio, Sameer K. Antani.

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
