## [Decision Letter · Decision Letter 0]

4 Sep 2020

PONE-D-20-22486

Analyzing inter-reader variability affecting deep ensemble learning for COVID-19 detection in chest radiographs

PLOS ONE

Dear Dr. Rajaraman,

Thank you for submitting your manuscript to PLOS ONE. After careful consideration, we feel that it has merit but does not fully meet PLOS ONE’s publication criteria as it currently stands. Therefore, we invite you to submit a revised version of the manuscript that addresses the points raised during the review process. If you decide to submit the revision, please provide point-to-point response to address all concerns from reviewers.

We look forward to receiving your revised manuscript.

Kind regards,

Yuankai Huo, Ph.D.

Academic Editor

PLOS ONE

Additional Editor Comments:

Thank you for submitting your manuscript to PLOS ONE. After careful consideration, we feel that it has merit but does not fully meet PLOS ONE’s publication criteria as it currently stands. Therefore, we invite you to submit a revised version of the manuscript that addresses the points raised during the review process. Please provide point-to-point response to address all concerns from reviewers.

Journal Requirements:

Reviewers' comments:

Reviewer's Responses to Questions

**Comments to the Author**

1. Is the manuscript technically sound, and do the data support the conclusions?

Reviewer #1: Partly

Reviewer #2: Yes

2. Has the statistical analysis been performed appropriately and rigorously? 

Reviewer #1: Yes

Reviewer #2: Yes

3. Have the authors made all data underlying the findings in their manuscript fully available?

Reviewer #1: Yes

Reviewer #2: Yes

4. Is the manuscript presented in an intelligible fashion and written in standard English?

Reviewer #1: Yes

Reviewer #2: Yes

5. Review Comments to the Author

Reviewer #1: Strength:

The authors focused on the COVID-19 topics, which is very important worldwide recently. This paper discussed several topics related to the deep learning and medical applications. As the authors state, it is first study to construct ensembles, perform ensemble-based disease ROI localization, and analyze inter- reader variability and algorithm performance for COVID-19 detection in CXRs.

Weakness:

1. There are several statements lack of refs, see the comments below for examples, the author should further read to if more statements need refs.

2. Some training details are missed, leading confusing reading.

For example, line 320. "a combination of binary cross-entropy and dice losses." can not lead to a "exact" expression. How do you combine that? How do you balance the multiple losses, if there weights between them?

Line 321, how you define the "best model", how you select the "best model"?

3. How you define the threshold of predicted probability, is it 0.5?

4. The figures are not friendly to read, please consider the PDF or eps format.

5. In my opinion, innovation is limited for technique perspective.

More Comments:

1. Please add refs at line 106 for the approaches, statement of line 122-124, etc.

2. In my opinion, please consider change the style "the authors of [#]" to others like "AuthorLastName et al. [#]".

3. Please add the loss function to the figures.

4. Please consider different weights between the cross-entropy and dice losses, and show a table comparison.

5. Please consider adding a figure to visualize all the training steps.

Reviewer #2: Summary and contributions:

The authors presented a systemic approach for chest x-ray COVID-19 classification model design/training and model analysis. Methodology-wise the authors proposed to use multi-stage modality-specific transfer learning and multi-model based ensemble learning which achieved good performance. In the post-analysis, the authors deployed CRM for classification model attention visualization and extensive statistical metrics for performance analysis. For the inter-reader variability study, the authors used STAPLE to compare the Kappa/Sensitivity/Specificity/PPV between readers and one model.

Strength:

1. The authors address the chest x-ray COVID-19 classification problem using modality-specific transfer learning and ensemble learning, achieving impressive performance.

2. The authors performed extensive statistical analysis to their models.

3. The author also studied the ROI variability between reader and model.

Weakness:

1. The technical contribution is limited. The modality-specific transfer learning and ensemble learning have been proposed before. The authors used them for COVID-19 application.

2. The dataset for evaluation is limited. Only 72 chest x-ray was used for evaluation and the disease severity range could be limited.

3. The paper is hard to read and contains too many subsections. The authors should consider to reorganize the paper by merging subsections to focus on two major parts. One is method (modality-specific transfer learning and ensemble learning) and another is analysis (performance analysis/ROI/inter-reader studies).

Additional Comments:

1. Line 680: In Fig 13, fine-tuned densenet gives false positive attention while in Table 9 densenet gives the best performance. What is the reason?

2. Line 874: Which program/model was used for the inter-reader study? Please clarify.

6. PLOS authors have the option to publish the peer review history of their article (what does this mean?). If published, this will include your full peer review and any attached files.

Reviewer #1: No

Reviewer #2: No

---

## [Author Response · Author response to Decision Letter 0]

20 Sep 2020

Reviewer#1: The authors focused on the COVID-19 topics, which is very important worldwide recently. This paper discussed several topics related to the deep learning and medical applications. As the authors state, it is first study to construct ensembles, perform ensemble-based disease ROI localization, and analyze inter- reader variability and algorithm performance for COVID-19 detection in CXRs.

Author response: We render our sincere thanks to the reviewer for the valuable comments and appreciation of our study. To the best of our knowledge and belief, we have addressed the reviewer’s concerns.

Reviewer#1, Concern # 1: There are several statements lack of refs, see the comments below for examples, the author should further read to if more statements need refs. Some training details are missed, leading confusing reading. For example, line 320. "a combination of binary cross-entropy and dice losses." can not lead to a "exact" expression. How do you combine that? How do you balance the multiple losses, if there weights between them?

Author response: We appreciate the reviewer’s concern in this regard. We regret the lack of clarity in the initial submission. We have combined the losses as shown below:

L_n=w_1 L_(BCE_n )+w_2 L_(DSC_n )

where L_(BCE_n ) is the binary cross-entropy loss, L_(DSC_n ) is the Dice loss, and n denotes the batch number. The losses are computed for each mini-batch. The final loss for the entire batch is determined by the mean of loss across all the mini-batches. The expression for L_(BCE_n ) and 〖L_DSC〗_nis given by:

L_(BCE_n )=-[t_n log⁡(y_n )+(1-t_n ) log⁡(1-y_n ) ]

〖L_DSC〗_n=1-(2∑▒〖t_n∙y_n 〗)/(∑▒t_n +∑▒y_n )

where t is the target and y is the output from the final layer. Here, we choose w1=w2=0.5. The model is trained and validated on patient-specific splits (80/20 train/validation split) of CXRs and their associated lung masks made available by Candemir & Antani [35]. We do not have the ground truth masks for the various CXR data collections used in this study. Hence, we were not able to evaluate the performance of the segmentation model for different combinations of weights for the losses. For this reason, we performed coarse segmentation by delineating the lung boundaries using the generated masks and cropped them to a bounding box containing the lung pixels so that the DL models train on the lung-specific ROI and avoid learning irrelevant features. We plan to experiment with combinations of weights in our future studies when the ground truth masks for the CXR collections used in the study are made publicly available.

Author action: We updated the manuscript with loss function equations and their explanation. These changes can be found on page 13, lines 292 – 306 of the revised manuscript.

Reviewer#1, Concern # 2: Line 321, how you define the "best model", how you select the "best model"?

Author response: Callbacks are used to store model weights after each epoch only when there is a reduction in the validation loss. This helps us select the “best model” at the end of the training phase.

Author action: We updated the manuscript with the above details. The changes can be found on page 13, lines 306 – 308 of the revised manuscript.

Reviewer#1, Concern # 3: How you define the threshold of predicted probability, is it 0.5?

Author response: We have used the default value of 0.5 as the discrimination threshold to convert the predicted probability into the class labels. This explanation can be found on page 13 , line 308 – 309 of the revised manuscript.

Reviewer#1, Concern # 4: The figures are not friendly to read, please consider the PDF or eps format. 

Author response: The original figures uploaded during submission have a resolution of 600 pixels per inch. They appear sharp, and we believe they are in compliance with PLOS ONE requirements. It is possible that the PDF formatting is reducing their clarity.

Reviewer#1, Concern # 5: In my opinion, innovation is limited for technique perspective.

Thanks for your comments in this regard. While there are a number of medical imaging CADx solutions that use DL approaches for disease detection including COVID-19, there are significant limitations in existing approaches related to data set size, scope, model architecture, and evaluation. Our innovative approach addresses these shortcomings and proposes novel analyses to meet the urgent demand for COVID-19 detection using CXRs through a systematic approach combining CXR modality-specific model pretraining, fine-tuning, and ensemble learning to improve COVID-19 detection in CXRs. We demonstrate that the ensemble-based ROI localization is better performing than standalone localization methods. Our empirical observations led to the conclusion that the classification accuracy of a model is not a sufficiently optimal measure to interpret its learned behavior. Localization studies are indispensable to understand the learned features, compare them with the expert knowledge for the problem under study, and provide comprehensive qualitative and quantitative measures of the learning capacity of the model. We also performed inter-reader variability analysis and program performance assessment by comparing them with a STAPLE-based estimated reference. This assessment highlighted the opportunity for improving performance through ensemble modifications, requisite parameter optimization, increased task-specific dataset size, and involving “truth” estimates from a larger number of expert collaborators. We believe that our manuscript establishes a paradigm for future research using ensemble-based classification, localization, and analyzing observer-variability in medical and other natural visual recognition tasks. The results proposed would be useful for developing robust models for tasks involving medical image classification and disease-specific ROI localization.

Reviewer#1, Concern # 6: Please add refs at line 106 for the approaches, statement of line 122-124, etc.

Author response: Agreed and thanks. We have included the following references per reviewer suggestions: 

1. Coronavirus disease (COVID-2019) situation reports. In: World Health Organization (WHO) Situation Reports. [Internet]. Jan 2020 [cited May 2020]. Available: https://www.who.int/emergencies/diseases/novel-coronavirus-2019/situation-reports

2. Rubin GD, Ryerson CJ, Haramati LB, Sverzellati N, Kanne JP, Raoof S, et al. The Role of Chest Imaging in Patient Management During the COVID-19 Pandemic: A Multinational Consensus Statement From the Fleischner Society [published online ahead of print, 2020 Apr 7]. Chest. 2020;158(1):106-116. doi:10.1016/j.chest.2020.04.003

3. ACR Recommendations for the use of Chest Radiography and Computed Tomography (CT) for Suspected COVID-19 Infection. In: Recommendations for Chest Radiography and CT for Suspected COVID19 Infection [Internet]. 11 Mar 2020 [cited 12 Mar 2020]. Available: https://www.acr.org/Advocacy-and-Economics/ACR-Position-Statements/Recommendations-for-Chest-Radiography-and-CT-for-Suspected-COVID19-Infection

11. Chowdhury AK, Tjondronegoro D, Chandran V, Trost SG. Ensemble Methods for Classification of Physical Activities from Wrist Accelerometry. Med Sci Sports Exerc. 2017;49(9):1965-1973. doi:10.1249/MSS.0000000000001291

Author action: The following references are included as shown:

A new pandemic, for example, may bias experts toward higher sensitivity, i.e. they will associate non-specific features with the new disorder because they lack experience with relevant disease manifestation in the image [1–3]. (Page 5 , line 113 – 116)

Ensemble learning methods including majority voting, averaging, weighted averaging, stacking, and blending seek to address these issues by combining predictions of multiple models and resulting in a better performance compared to that of any individual constituent model [11]. (Page 5 , line 100 – 102)

Reviewer#1, Concern # 7: In my opinion, please consider change the style "the authors of [#]" to others like "AuthorLastName et al. [#]".

Author response: Agreed and thanks. We have changed the style of referencing per reviewer suggestions. This was done throughout the revised manuscript.

Reviewer#1, Concern # 8: Please add the loss function to the figures.

Author response: Agreed. We thank the reviewer for the suggestion. We have included the performance curves for the custom U-Net model in the revised manuscript. Please refer to modified Fig 1. 

Author action: The performance curves for the custom U-Net model is added to Fig 1. 

Reviewer#1, Concern # 9: Please consider different weights between the cross-entropy and dice losses, and show a table comparison.

Author response: Thanks. We wish to reiterate our response to the reviewer concern #1 to this end. We do not have the ground truth masks for CXR data collections used in this study. Hence, we were not able to evaluate the performance of the segmentation model for a different combination of weights for the losses. For this reason, we performed coarse segmentation by delineating the lung boundaries using the generated masks and cropped them to a bounding box containing the lung pixels. We plan to experiment with the combination of weights in our future studies when the ground truth masks for the CXR collections are made publicly available.

Reviewer#1, Concern # 10: Please consider adding a figure to visualize all the training steps. 

Author response: Agreed and thanks. We have already included an image/graphical abstract of our training steps with the initial submission. In this revised version, we have added step numbers in the text and also to Fig. 3 to help relate them. We have shown the revised text and figure below.

Author action: The following changes are made to the manuscript text and Fig 3. (Page 14 , line 324 – 333)

The steps in training that follow segmentation are shown in Fig 3. First (1), the images are preprocessed to remove irrelevant features by cropping the lung ROI. The cropped images are used for model training and evaluation. We perform repeated CXR-specific pretraining in transferring modality-specific knowledge that is fine-tuned toward detecting COVID-19 viral manifestations in CXRs. To do this, in the next training step (2) the CNNs are trained on a large collection of CXRs to separate normals from those showing abnormalities of any type. Next, (3) we retrain the models from the previous step, focusing on separating CXRs showing bacterial pneumonia or non-COVID-19 viral pneumonia from normals. Next, (4) we fine-tune the models from the previous step toward the specific separation of CXRs showing COVID-19 pneumonia from normals. Finally (5) the learned features from this phase of training become parts of the ensembles developed to optimize the detection of COVID-19 pneumonitis from CXRs.

Reviewer#2: The authors presented a systemic approach for chest x-ray COVID-19 classification model design/training and model analysis. Methodology-wise the authors proposed to use multi-stage modality-specific transfer learning and multi-model based ensemble learning which achieved good performance. In the post-analysis, the authors deployed CRM for classification model attention visualization and extensive statistical metrics for performance analysis. For the inter-reader variability study, the authors used STAPLE to compare the Kappa/Sensitivity/Specificity/PPV between readers and one model.

 The authors address the chest x-ray COVID-19 classification problem using modality-specific transfer learning and ensemble learning, achieving impressive performance.

 The authors performed extensive statistical analysis to their models.

 The author also studied the ROI variability between reader and model.

Author response: We thank the reviewer for the appreciation and insightful comments on this study. To the best of our knowledge and belief, we have addressed the concerns of the reviewer to make the manuscript suitable for a possible publication. 

Reviewer#2, Concern # 1: The technical contribution is limited. The modality-specific transfer learning and ensemble learning have been proposed before. The authors used them for COVID-19 application.

Author response: We thank the reviewer for his comments in this regard. While there are a number of medical imaging CADx solutions that use DL approaches for disease detection including COVID-19, there are significant limitations in existing approaches related to data set size, scope, model architecture, and evaluation. We address these shortcomings and propose novel analyses to meet the urgent demand for COVID-19 detection using CXRs. This study is superior to our previous publication in several aspects: The current study proposes the benefits of a systematic approach combining CXR modality-specific model pretraining, fine-tuning, and ensemble learning to improve COVID-19 detection in CXRs. We demonstrate that the ensemble-based region of interest (ROI) localization is better performing than standalone localization methods. Our empirical observations led to the conclusion that the classification accuracy of a model is not an optimal measure to interpret its learned behavior. Localization studies are indispensable to understand the learned features and compare them to the expert knowledge for the problem under study. We provide comprehensive qualitative and quantitative measures of the learning capacity of the model. We also performed inter-reader variability analysis and program performance assessment by comparing them with a STAPLE-based estimated reference. This assessment highlighted the opportunity for improving performance through ensemble modifications, requisite parameter optimization, increased task-specific dataset size, and involving “truth” estimates from a larger number of expert collaborators. We believe that our manuscript establishes a paradigm for future research using ensemble-based classification, localization, and analyzing observer-variability in medical and other natural visual recognition tasks. The results proposed would be useful for developing robust models for tasks involving medical image classification and disease-specific ROI localization. 

Reviewer#2, Concern # 2: The dataset for evaluation is limited. Only 72 chest x-ray was used for evaluation and the disease severity range could be limited.

Author response: We agree that the number of publicly available CXRs showing COVID-19 manifestations are limited at present. In spite of limited data availability, however, we empirically demonstrate a stage-wise, systematic approach for improving classification and ROI localization performance through modality-specific transfer learning. This, in turn, helped learn the common characteristics of the source and target modalities and lead to a better initialization of model parameters and faster convergence. This further reduced computational demand, improved efficiency, and increased the opportunity for potential successful deployment. We proposed the benefits from performing ensemble learning, particularly under sparse data availability as is the case in our study, which combined the predictions of multiple models and resulted in a better performance compared to that of any individual constituent model. This is the first study to propose ensemble-based ROI localization, particularly applied to COVID-19 detection in CXRs. Such a localization method helped in compensating for localization errors and missed ROIs by combining and averaging the individual class-relevance maps. Our empirical evaluations show that ensemble localization demonstrated superior IoU and mAP scores and they significantly outperform ROI localization by individual CNN models. Ensemble-based localization demonstrated superior performance under current conditions of sparse data availability. This study establishes a paradigm for developing robust, ensemble-based models for tasks involving medical image classification and disease-specific ROI localization, particularly under circumstances of limited data availability. We sincerely believe that this innovative approach would only result in improved performance and generalization with more data available in the future. 

Reviewer#2, Concern # 3: The paper is hard to read and contains too many subsections. The authors should consider to reorganize the paper by merging subsections to focus on two major parts. One is method (modality-specific transfer learning and ensemble learning) and another is analysis (performance analysis/ROI/inter-reader studies).

Author response: Agreed. We have revised the manuscript structure per reviewer suggestions. The manuscript structure is reorganized by merging the sub-sections under “Introduction” and “Materials and methods” Sections into two major parts: a) Modality-specific transfer learning and ensemble learning, and b) ROI localization, observer variability, and statistical analysis. We moved the description pertaining to the STAPLE algorithm and the performance measures used to assess observer variability and algorithmic performance to a supplement file named “S1_File.pdf”. We also merged sub-sections in the “Results” section to improve readability.

Reviewer#2, Concern # 4: Line 680: In Fig 13, fine-tuned densenet gives false positive attention while in Table 9 densenet gives the best performance. What is the reason?

Author response: We are happy to clarify on the reviewer’s query in this regard. As observed from Table 9, the modality-specific pretrained/finetuned ResNet-18 demonstrated superior performance (Acc: 0.8958; AUC: 0.9477) as compared to other fine-tuned models. The image pairs in Fig. 13 strongly convey the point that the importance of fine tuning is typically validated by the ability of these models to localize COVID-19 manifestations correctly. When fine tuning is absent, the disease-specific ROI localization tends to be poor. We agree that the DenseNet-121, inspite of delivering good classification performance, only next to ResNet-18 and MobileNet-V2 finetuned models, is showing from false attention. However, this is specific to this CXR image; the pattern is likely not to repeat for all CXR images. We appreciate that the DenseNet-121 example is confusing in this figure and have removed it and replaced with ResNet-18-based baseline and fine-tuned model localization. However, this doesn’t change the fact that the quantitative accuracy as seen in the AUC does not always match localization quality. The CRM-based localization studies helped in identifying a) whether the trained model is classifying the CXRs to their respective classes based on the task-specific features and not the surrounding context, b) gaining a clear understanding of the learned behavior of the model, and c) comparing them to the expert knowledge for the problem under study. 

Reviewer#2, Concern # 5: Line 874: Which program/model was used for the inter-reader study? Please clarify.

Author response: Thanks. The Simultaneous Truth and Performance Level Estimation (STAPLE) algorithm is used to generate a reference consensus annotation from the set of radiologists’ annotations. This is compared with individual radiologist annotations and the predicted disease ROI by model ensembles to provide a measure of inter-reader variability and algorithm performance. The metrics including Kappa statistic, Sensitivity, Specificity, PPV, and NPV are used to analyze the variability between the reference annotation, radiologists’ annotations, and predicted masks.

---

## [Decision Letter · Decision Letter 1]

21 Oct 2020

PONE-D-20-22486R1

Analyzing inter-reader variability affecting deep ensemble learning for COVID-19 detection in chest radiographs

PLOS ONE

Dear Dr. Rajaraman,

Thank you for submitting your manuscript to PLOS ONE. After careful consideration, we feel that it has merit but does not fully meet PLOS ONE’s publication criteria as it currently stands. Therefore, we invite you to submit a revised version of the manuscript that addresses the points raised during the review process.

We look forward to receiving your revised manuscript.

Kind regards,

Yuankai Huo, Ph.D.

Academic Editor

PLOS ONE

Additional Editor Comments (if provided):

This paper is conditional accepted once the minor issues are addressed.

Reviewers' comments:

Reviewer's Responses to Questions

**Comments to the Author**

1. If the authors have adequately addressed your comments raised in a previous round of review and you feel that this manuscript is now acceptable for publication, you may indicate that here to bypass the “Comments to the Author” section, enter your conflict of interest statement in the “Confidential to Editor” section, and submit your "Accept" recommendation.

Reviewer #1: All comments have been addressed

Reviewer #2: All comments have been addressed

2. Is the manuscript technically sound, and do the data support the conclusions?

Reviewer #1: Yes

Reviewer #2: Partly

3. Has the statistical analysis been performed appropriately and rigorously? 

Reviewer #1: Yes

Reviewer #2: Yes

4. Have the authors made all data underlying the findings in their manuscript fully available?

Reviewer #1: Yes

Reviewer #2: Yes

5. Is the manuscript presented in an intelligible fashion and written in standard English?

Reviewer #1: Yes

Reviewer #2: Yes

6. Review Comments to the Author

Reviewer #1: Generally, my comments have been addressed. Please provide a demographic table of the patients if possible.

Reviewer #2: The authors satisfactorily addressed my concerns and revised the manuscript correspondingly. The revised paper is still a very long paper, I would recommend to move some non-essential content to supplementary or simplify the content so reader can concentrate on the main ideas (such as the data distribution tables could be merged to one if possible).

7. PLOS authors have the option to publish the peer review history of their article (what does this mean?). If published, this will include your full peer review and any attached files.

Reviewer #1: No

Reviewer #2: No

---

## [Author Response · Author response to Decision Letter 1]

22 Oct 2020

Reviewer#1: Generally, my comments have been addressed. Please provide a demographic table of the patients if possible.

Author response: We render our sincere thanks to the reviewer for the valuable comments and appreciation of our study. As recommended, we have included the demographic information provided by the data providers for the various datasets used in this study.

Reviewer#2: The authors satisfactorily addressed my concerns and revised the manuscript correspondingly. The revised paper is still a very long paper, I would recommend to move some non-essential content to supplementary or simplify the content so reader can concentrate on the main ideas (such as the data distribution tables could be merged to one if possible).

Author response: We thank the reviewer for the appreciation and insightful comments on this study. As recommended, we have made the following changes to the revised manuscript: 

1. The details pertaining to the datasets and their distribution, used in various stages of learning, are merged into a single table (Table 2).

2. Per reviewer suggestions, the following information have been moved to the supplementary material (S1 File.pdf) to help the readers concentrate on the main ideas:

1. Inter-reader variability analysis using STAPLE algorithm (Section A of the supplement S1 File)

2. Class-selective relevance (CRM) visualization (Section B of the supplement S1 File)

3. Table showing empirically determined feature extraction layers and its related discussion (Section C of the supplement S1 File)

4. t-SNE visualization of feature embedding (Section D of the supplement S1 File)

5. P-R curves for the top-performing individual models (Section E of the supplement S1 File)

6. P-R curves for top-N ensemble CRMs (Section F of the supplement S1 File)

---

## [Editor Report · Decision Letter 2]

2 Nov 2020

Analyzing inter-reader variability affecting deep ensemble learning for COVID-19 detection in chest radiographs

PONE-D-20-22486R2

Dear Dr. Rajaraman,

We’re pleased to inform you that your manuscript has been judged scientifically suitable for publication and will be formally accepted for publication once it meets all outstanding technical requirements.

Kind regards,

Yuankai Huo, Ph.D.

Academic Editor

PLOS ONE
---

## [Editor Report · Acceptance letter]

4 Nov 2020

PONE-D-20-22486R2 

Analyzing inter-reader variability affecting deep ensemble learning for COVID-19 detection in chest radiographs 

Dear Dr. Rajaraman:

I'm pleased to inform you that your manuscript has been deemed suitable for publication in PLOS ONE. Congratulations! Your manuscript is now with our production department. 

Kind regards, 

on behalf of

Dr. Yuankai Huo 

Academic Editor

PLOS ONE